# Single-keratinocyte transcriptomic analyses identify different clonal types and proliferative potential mediated by FOXM1 in human epidermal stem cells

Elena Enzo[1,10], Alessia Secone Seconetti[1,2,10], Mattia Forcato [3], Elena Tenedini[4], Maria Pia Polito [1], Irene Sala[1], Sonia Carulli[2], Roberta Contin[1,9], Clelia Peano [5,6], Enrico Tagliafico[4,7,8], Silvio Bicciato [3], Sergio Bondanza[2] & Michele De Luca [1✉]

Autologous epidermal cultures restore a functional epidermis on burned patients. Transgenic epidermal grafts do so also in genetic skin diseases such as Junctional Epidermolysis Bullosa. Clinical success strictly requires an adequate number of epidermal stem cells, detected as holoclone-forming cells, which can be only partially distinguished from the other clonogenic keratinocytes and cannot be prospectively isolated. Here we report that single-cell transcriptome analysis of primary human epidermal cultures identifies categories of genes clearly distinguishing the different keratinocyte clonal types, which are hierarchically organized along a continuous, mainly linear trajectory showing that stem cells sequentially generate progenitors producing terminally differentiated cells. Holoclone-forming cells display stem cell hallmarks as genes regulating DNA repair, chromosome segregation, spindle organization and telomerase activity. Finally, we identify *FOXM1* as a YAP-dependent key regulator of epidermal stem cells. These findings improve criteria for measuring stem cells in epidermal cultures, which is an essential feature of the graft.

[1] Centre for Regenerative Medicine "Stefano Ferrari", University of Modena and Reggio Emilia, Modena, Italy. [2] Holostem Terapie Avanzate, s.r.l, Modena, Italy. [3] Department of Life Sciences, University of Modena and Reggio Emilia, Modena, Italy. [4] Department of Laboratory Medicine and Pathology, Diagnostic hematology and Clinical, Genomics Unit, Modena University Hospital, Modena, Italy. [5] Genomic Unit, IRCSS, Humanitas Clinical and Research Center, Rozzano, Milan, Italy. [6] Institute of Genetic and Biomedical Research, UoS Milan, National Research Council, Rozzano, Italy. [7] Department of Medical and Surgical Sciences, University of Modena and Reggio Emilia, Modena, Italy. [8] Centre for Genome Research, University of Modena and Reggio Emilia, Modena, Italy. [9] Present address: Clinical Sampling & Alliances, AstraZeneca, Cambridge, UK. [10] These authors contributed equally: Elena Enzo, Alessia Secone Seconetti. ✉email: michele.deluca@unimore.it

The mammalian interfollicular epidermis contains clonogenic keratinocytes that reside in the basal layer, constitute the epidermal proliferative compartment, and firmly adhere to the basement membrane through hemidesmosomes. Nature, properties, and roles of clonogenic keratinocytes in governing continuous renewal and timely repair of mammalian epidermis have been debated[1]. In mice, lineage tracing pointed to either a homogeneous population of equipotent progenitors producing suprabasal keratinocytes or hierarchically organized stem cells and transient amplifying (TA) progenitors, the latter eventually generating terminally differentiated cells[2,3]. In humans, the heterogeneity of clonogenic keratinocytes was first unveiled by the discovery that such cells can initiate at least three types of clones, referred to as holoclones, meroclones, and paraclones[4], all of which proliferate. However, only the holoclone-forming cell has hallmarks of a stem cell, being endowed with long-term proliferative potential and self-renewal capacity[5]. Clonal conversion, that is the transition from holoclones to meroclones to paraclones, precedes the onset of terminal differentiation[4–7].

Compelling, yet indirect, evidence of functional heterogeneity of human clonogenic keratinocytes came from their clinical use in regenerative medicine. Cultured epithelial sheets containing all types of clonogenic keratinocytes are successfully used for the treatment of massive full-thickness skin burns and ocular burns with limbal stem cell deficiency[8,9]. Permanent regeneration of both epidermis and corneal epithelium strictly requires a defined number of holoclone-forming cells contained in the culture[10,11].

Formal evidence of holoclone-forming cells being stem cells able to permanently sustain the human epidermis came from the life-saving use of transgenic epidermal cultures for combined cell and gene therapy of junctional epidermolysis bullosa (JEB), a severe genetic skin disease. Virtually the entire epidermis of a 7-year-old child suffering from a devastating form of JEB with very poor prognosis has been permanently restored by means of such cultures[5]. Using proviral integrations as clonal genetic marks, clonal tracing performed on the regenerated transgenic epidermis has formally shown that the human epidermis is sustained solely by self-renewing holoclone-forming cells. They continuously generate pools of meroclones and paraclones, which behave as TA progenitors, persist for various periods of time, replenish differentiated cells, and play a crucial role both in the engraftment of epidermal cultures and in epidermal regeneration during wound healing[5]. Thus, at least in humans, these findings clearly pointed to a model envisaging long-lived stem cells generating short-lived progenitors eventually producing terminally differentiated cells.

Despite decades of clinical application of epithelial cultures, the cumbersome and lengthy procedures required for keratinocyte clonal analyses hampered the thorough characterization of molecular pathways governing self-renewal in holoclones and distinguishing them from meroclones (and paraclones). An important step toward the molecular identification of human keratinocyte clonal types came from the discovery of p63 as a key transcription factor underpinning the proliferative potential of epithelial stem cells[12–15]. In addition, it has recently been shown that nuclear YAP is a key determinant of human holoclones and its interplay with p63 is essential for sustaining their self-renewal and proliferative/regenerative capacity[16].

Here we show that genome-wide transcriptome analysis performed on single clonogenic primary human epidermal keratinocytes identifies categories of genes clearly distinguishing the different clonal types and discover the role of FOXM1 as a YAP-dependent key regulator of normal and adhesion-defective epidermal stem cells.

## Results

**Microarray analysis of different epidermal clonal types.** While paraclones can be identified based on their peculiar morphology[4], holoclones and meroclones cannot be distinguished based on their growth rate and behavior and shape and size of the colonies (Supplementary Fig. 1a). To explore whether holoclones, meroclones, and paraclones activate different transcriptional programs, we performed genome-wide microarray gene expression profiling of 60 clones (20 holoclones, 29 meroclones, and 11 paraclones) isolated from six different primary cultures (K5, K18, K22, K38, K42, and K49) (Fig. 1a, 1) ("Methods"). Unsupervised principal component analysis (PCA) showed that holoclones are bundled in a rather homogeneous cluster (Fig. 1b, red dots) regardless of the strain analyzed (Supplementary Fig. 1b).

Differential expression analysis unveiled 551 genes upregulated in holoclones as compared to meroclones (false discovery rate (FDR) ≤ 5% and fold change ≥ 1.5; Fig. 1c, blue circle) and 1480 genes upregulated in holoclones as compared to paraclones (FDR ≤ 5% and fold change ≥ 1.5; Fig. 1c, gray circle). A total of 526 genes (i.e., 95% of those 551 genes) were found to be upregulated when comparing holoclones to either meroclones or paraclones, suggesting a transcriptional signature progressively distinguishing holoclones from the two other epidermal clonal types. We defined these 526 genes as *holoclone signature* (Supplementary Data 1). We selected some of the more differentially expressed genes, not simply related to the cell cycle, contained in the holoclone signature (i.e., *ANLN*, *AURKB*, *CCNA2*, *CKAP2L*, *FOXM1*, *HMGB2*, and *LMNB1*) and confirmed, by quantitative reverse transcription PCR (qRT-PCR), that their expression was indeed significantly higher in holoclones as compared to meroclones (Fig. 1d).

Functional annotation using Gene Ontology analysis revealed that genes upregulated in holoclones were associated to cell cycle, DNA repair, chromosome segregation, and spindle organization (Fig. 1e and Supplementary Fig. 1c). Gene Set Enrichment Analysis (GSEA) confirmed that signatures related to DNA repair, telomerase activity, and cell cycle control were highly up regulated in holoclones as compared to the other clonal types (Fig. 1f and Supplementary Data 2). Intriguingly, GSEA unveiled that pathways usually detected in human embryonic stem cells, as *ES1*, *ES2*, *Tumorigenic stem*, *ES-like* (Fig. 1f) were significantly activated in holoclones when compared to both meroclones (Fig. 1f, left) and paraclones (Fig. 1f, right). It is worth noting that also YAP target genes are significantly activated in holoclones (Fig. 1f).

We found that 477 genes were downregulated in holoclones as compared to meroclones (FDR ≤ 5% and fold change ≤ −1.5; Fig. 1g, blue circle) and 352 of these genes (68%) were in common with the 1160 genes downregulated in holoclones as compared to paraclones (Fig. 1g, gray circle). Genes downregulated in holoclones and progressively increasing in meroclones and paraclones were linked to epidermal differentiation, response to wounding and other organisms, including bacteria (Supplementary Fig. 1d).

These data show that holoclones and meroclones, despite being indistinguishable in vitro, possess distinct molecular signatures and suggest a progressive change of gene expression marking the transition from holoclones to meroclones to paraclones along a keratinocyte differentiation route.

**Single-cell RNA-seq of different colony-forming keratinocytes.** Since microarray analyses were performed on clones generated by single cells cultivated for 1 week ("Methods" and Fig. 1a), they cannot provide conclusive information on the original clone-founding cell. But, while holoclones contain cells able to initiate

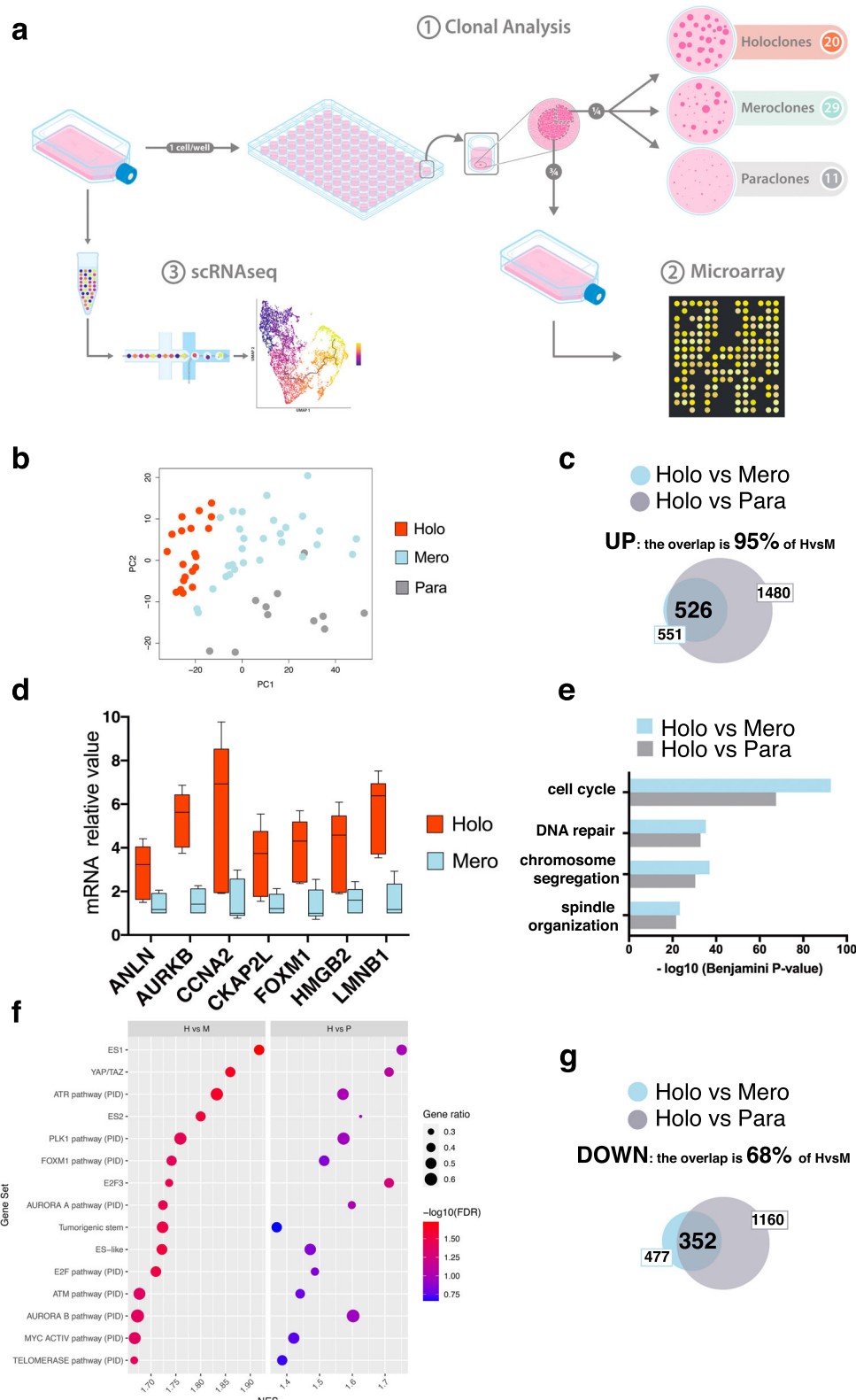

all clonal types upon sub-cultivation, holoclone-forming cells are not contained in meroclones and paraclones[6]. Thus, holoclone-forming cells must express the holoclone-specific genes distinguishing them from the other clonal types.

To define the transcriptional profile of holoclone-forming cells, we thus performed single-cell RNA-seq analyses on human keratinocytes, using the *holoclone signature* as a readout. We initially profiled

the expression of 3.367 and 3.978 cells obtained from two different sub-confluent primary epidermal cultures (K82 and K86) (Fig. 2a, t₁). Clustering identified nine cell types within the integrated data regressed for cell cycle, in order to avoid any bias linked to it. Clusters generated by lethally irradiated 3T3-J2 feeder cells and/or human fibroblasts (Fig. 2a, left panel, F1-2) have been identified by vimentin expression levels (Supplementary Fig. 2a) and not considered for the

**Fig. 1 Gene expression profile of human epidermal clonal types. a** From left to right: sub-confluent cultures were trypsinized, serially diluted, and inoculated (1 cell per well) onto 96-multiwell plates containing irradiated 3T3-J2 cells. After 7 days of cultivation, single clones were identified under an inverted microscope, trypsinized, transferred to two plates, and cultivated. One plate (one-quarter of the clone) was fixed 12 days later and stained with rhodamine B for the classification of clonal type (1, clonal analysis), which was determined as described in "Methods". The second plate (three-quarters of the clone) was used for microarray analysis (2). The same sub-confluent cultures were used for single-cell RNA sequencing using 10X Genomics platform (3, scRNA-seq). **b** Principal component analysis (PCA). Each dot represents a different clone ($n = 60$). Holoclones, meroclones, and paraclones are identified with red, light blue, and gray dots, respectively. **c** Venn diagram showing the overlap between the genes significantly upregulated (FDR ≤ 5% and fold change ≥ 1.5) in holoclones as compared to meroclones (blue circle) or paraclones (gray circle). **d** qRT-PCR quantification of the mRNA levels of some of the genes comprised in the holoclone signature (*ANLN, AURKB, CCNA2, CKAP2L, FOXM1, HMGB2,* and *LMNB1*) ($n = 5$ holoclones and 5 meroclones). Expression levels were normalized per *GAPDH* and are given relative to one meroclone (arbitrarily set to 1). Holoclones and meroclones are identified with red and light blue columns, respectively. Median and min to max values displayed. **e** Gene ontology (GO) analysis of the genes upregulated (see panel **c**) in holoclones as compared to meroclones (blue bars) and to paraclones (gray bars). *P* values are calculated with one-sided Fisher's Exact test and corrected for multiple tests with the Benjamini–Hochberg method. Histograms represent –log10 of the Benjamini–Hochberg corrected *p* value. **f** Bubble plot showing results of gene set enrichment analysis (GSEA) on microarray data obtained from the 60 clones of panel **b**. The normalized enrichment score (NES) is indicated on the *x*-axis; the dot size indicates the fraction of genes contributing to the leading-edge subset within the gene set. Dots are color-coded based on the enrichment FDR. The top 15 significant terms upregulated in holoclones as compared to both meroclones and paraclones are shown. **g** Venn diagram showing the overlap between the genes significantly downregulated (FDR ≤ 5% and fold change ≤ −1.5) in holoclones as compared to meroclones (blue circle) or paraclones (gray circle).

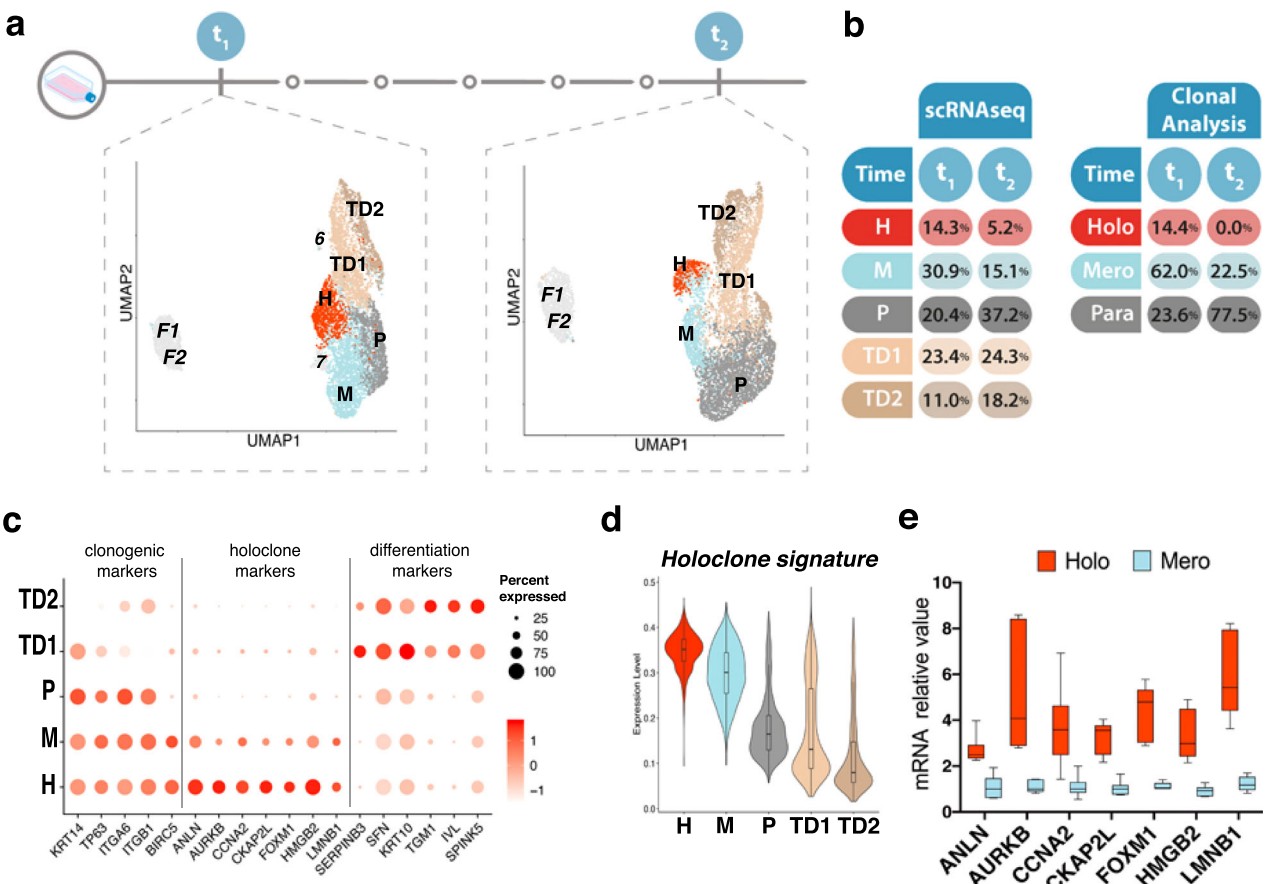

**Fig. 2 Single-cell RNA-seq analysis of human primary keratinocytes. a** Uniform manifold approximation and projection (UMAP) of the single-cell RNA-seq dataset comprising two keratinocyte primary cultures, both harvested at $t_1$ (left) and after serial cultivation ($t_2$, right; UMAP has been horizontally flipped for graphical purpose). Keratinocyte clusters (H, M, P, TD1, TD2) are colored according to cluster identity. Fibroblasts and low-quality keratinocytes (F1, F2, 6, 7) are shown in gray. **b** Table showing the percentage of cells contained in each cluster after scRNA-seq identified at $t_1$ and $t_2$ (left) and the percentage of each clonal type found after clonal analysis in $t_1$ or $t_2$ (right). **c** DotPlot showing expression of clonogenic, holoclone, and differentiation markers used to annotate the five keratinocyte clusters identified at $t_1$. **d** Violin plot showing the expression of the Holoclone signature in the keratinocyte clusters identified at $t_1$ ($n = 6273$ cells). In boxplots, lines in the middle of boxes correspond to median values. Lower and upper hinges correspond to the first and third quartiles, the upper whisker extends from the hinge to the largest value no further than 1.5 × IQR (inter-quartile range) from the hinge. The lower whisker extends from the hinge to the smallest value at most 1.5 × IQR of the hinge. **e** qRT-PCR quantification of the mRNA levels of some of the genes comprised in the holoclone signature (*ANLN, AURKB, CCNA2, CKAP2L, FOXM1, HMGB2,* and *LMNB1*) ($n = 6$ holoclones and 6 meroclones) on clones generated from K86 and K82 primary cultures. Expression levels were normalized per *GAPDH* and given relative to one meroclone (arbitrarily set to 1). Holoclones and meroclones are identified with red and light blue columns, respectively. Median and min to max values displayed.

remaining analyses (see "Methods" for details). Two small keratinocyte clusters (Fig. 2a, t₁, 6 and 7) were excluded from the analysis, as cluster 6 (accounting for 2.4% of total cells) contained cells with very low amounts of mRNA (Supplementary Fig. 2b) and cluster 7 (0.9%) showed high levels of stress-response-related genes (Supplementary Fig. 2c).

Five keratinocyte clusters (Fig. 2a, t₁ and Fig. 2b, sc-RNA-seq) were further analyzed based on expression of already known clonogenic and differentiation markers and our defined *holoclones signature*. Human clonogenic keratinocytes adhere to the basal lamina mainly through α6β4 integrins[17,18] are enriched in β1 integrins[19] and express the transcription factor p63[20], the transcriptional co-activator YAP, and survivin (a YAP target gene encoded by *BIRC5*)[16,21]. All the above genes identified clusters H, M, and P (14.3%, 30.9%, and 20.4%, respectively; Fig. 2c, clonogenic markers), suggesting that they were generated by clonogenic keratinocytes. Strikingly, the 526 genes of the holoclone signature (Fig. 1c) were strongly upregulated in cluster H and progressively decreased in clusters M and P (Fig. 2d). As expected (see also Fig. 1e, f), a comparable percentage of cells in G2/M was detected in clusters H and M (Supplementary Fig. 2d). Indeed, these findings are consistent with the notion that the vast majority (over 95%) of keratinocytes forming an holoclone is able to re-initiate daughter colonies able to proliferate and self-renew, while meroclones and paraclones are formed by keratinocytes that, although still proliferating within the colony, progressively lose their capacity to re-form new growing colonies[4,6,22,23]. As with epidermal strains shown in Fig. 1e, clonal analysis on K82 and K86 cultures confirmed that *ANLN*, *AURKB*, *CCNA2*, *CKAP2L*, *FOXM1*, *HMGB2*, and *LMNB1* were upregulated in holoclones and progressively decreased in meroclones and paraclones (Fig. 2e). As expected, these genes were highly expressed in cluster H and progressively decreased in clusters M and P (Fig. 2c, holoclone markers). In agreement with expression profiling of the clonal types (Fig. 1f), GSEA analysis showed that holoclone-forming cells were enriched in signatures related to DNA repair, cell cycle control, human stem cells, and YAP activity when compared to both meroclones and paraclones-forming cells (Supplementary Fig. 2e and Supplementary Data 2). The clusters that we designate as TD1 and TD2 (23.4% and 11%, respectively) express high levels of markers of terminal differentiation, such as *SERPINB3*, *SFN*, *KRT10*, *TGM1*, *IVL*, and *SPINK5* (Fig. 2c, differentiation markers).

Cluster dimensions (Fig. 2b, sc-RNA-seq, t₁) are consistent with the percentage of clonal types identified by clonal analyses of the original primary cultures (Fig. 2b, clonal analysis, t₁). Each cluster was equally represented in K82 and K86 cells, in line with the high reproducibility of human keratinocyte primary cultures (Supplementary Fig. 2f).

To strengthen the notion that cluster H contains holoclone-forming cells, K82 and K86 primary keratinocyte cultures were serially cultivated for five passages to induce clonal conversion and holoclones exhaustion. Cultures were passaged at full confluence to foster clonal conversion. As expected, clonal analysis performed on such cultures clearly showed absence of holoclones, a strong decrease of meroclones and a remarkable increase of paraclones (Fig. 2b, clonal analysis, t₂). Accordingly, RNA-seq analyses of 4292 and 3576 cells from serially cultured t₂ K82 and K86, respectively, unveiled a drastic decrease of clusters H and M and a strong increase of clusters P and TD1-2 (Fig. 2a, t₂ and Fig. 2b, scRNA-seq, t₂). The notion that cluster H can still be identified in the absence of detectable holoclones (Fig. 2a, b) suggests that either cluster H contains, but does not entirely consist of holoclone-forming cells, or arbitrary criteria for holoclone classification (≤5% of aborted colonies)[4] are too stringent and some meroclones (generating a low amount of aborted colonies) could indeed behave as stem cells.

Monocle3 computational analysis (able to determine dynamic processes of cells based on changes of their gene expression) organized keratinocytes identified at t₁ along a continuous, mainly linear trajectory showing, along with pseudotemporal ordering, that the stem cell cluster H sequentially generated TA progenitors (clusters M and P) eventually producing terminally differentiated keratinocytes (TD1 and TD2) (Fig. 3a, b). Identity and hierarchy of keratinocyte clusters were further confirmed by kinetics plot showing relative expression of clonogenic, holoclone, and differentiation marker genes across developmental pseudo-time (Fig. 3c).

## FOXM1 underpins holoclone-forming cells and acts downstream of YAP

Among the gene sets enriched in both holoclones and holoclone-forming cells, we fastened on FOXM1, a transcription factor member of the forkhead box family[24,25]. The FOXM1 pathway is among the gene sets enriched in holoclones both in microarray (Fig. 1f) and in scRNA-seq data (Supplementary Fig. 2e). Moreover, beside its role in driving G2/M transition in embryonic (and cancer) stem cells[26,27], FOXM1 controls self-renewal of both neural[28] and hematopoietic[29] stem cells, regeneration of striate muscles[30] and long-term maintenance of bronchiolar epithelium[31].

*FOXM1* is comprised in the holoclone signature (Figs. 1d and 2e), is highly expressed in holoclone-forming cells and barely detectable in meroclone-forming cells (Fig. 2c). This notion was confirmed by Western analysis (Fig. 4a) showing that FOXM1 was abundantly expressed in holoclones, barely detectable in meroclones and undetectable in paraclones. As expected, clonal conversion was also marked by a progressive decrease of p63 and survivin and a progressive increase of 14-3-3σ[16,20,22]. Integrins β1 and β4 were expressed at similar levels in holoclones and meroclones and strongly decreased in paraclones (Fig. 4a).

In analyzing the effect of FOXM1 on epidermal stem cells, it should be considered that, among all p63 isoforms, ΔNp63α (hereafter referred to as p63) is a key determinant of the proliferative potential of stem cells of all mammalian stratified epithelia[13]. Furthermore, although YAP is expressed in all epidermal clonal types, nuclear, transcriptionally active YAP is highly and selectively expressed in holoclones, barely detectable in meroclones and undetectable in paraclones[16].

Short-term (5 days) depletion of FOXM1 on different strains of primary keratinocytes by two independent siRNAs (Supplementary Fig. 3a, b) did not significantly decrease the number of cells (Supplementary Fig. 3c), nor the number of mitotic figures (Supplementary Fig. 3d), nor the expression of p63 (Supplementary Fig. 3b), nor the distribution of cells in the cell cycle phases by Edu/FxCycle incorporation (Supplementary Fig. 3f, g) suggesting that keratinocyte proliferation per se was not significantly impaired by FOXM1 depletion. Of note, short-term FOXM1 downregulation did not affect YAP (and its *CTGF* target) expression nor its nuclear localization (Supplementary Fig. 3a, e).

To investigate the long-term effect of FOXM1 depletion, human primary keratinocytes were transduced with an inducible lentiviral vector expressing either a control shRNA (shC) or two independent shRNA against *FOXM1* under a doxycycline-inducible Tet-promoter. At passage 3 after transduction, FOXM1, p63, and survivin were virtually undetectable (Fig. 4b), further suggesting a potential role of FOXM1 in sustaining epidermal stem cells. We thus analyzed 346 clones isolated from three strains of human primary keratinocytes (K5, K52, and K71) transduced with shC, shFOXM1#1 (shF1), and shFOXM1#5 (shF5) (see "Methods"). Strikingly, ablation of FOXM1 induced the selective disappearance (strains K5 and K71) or a decrease

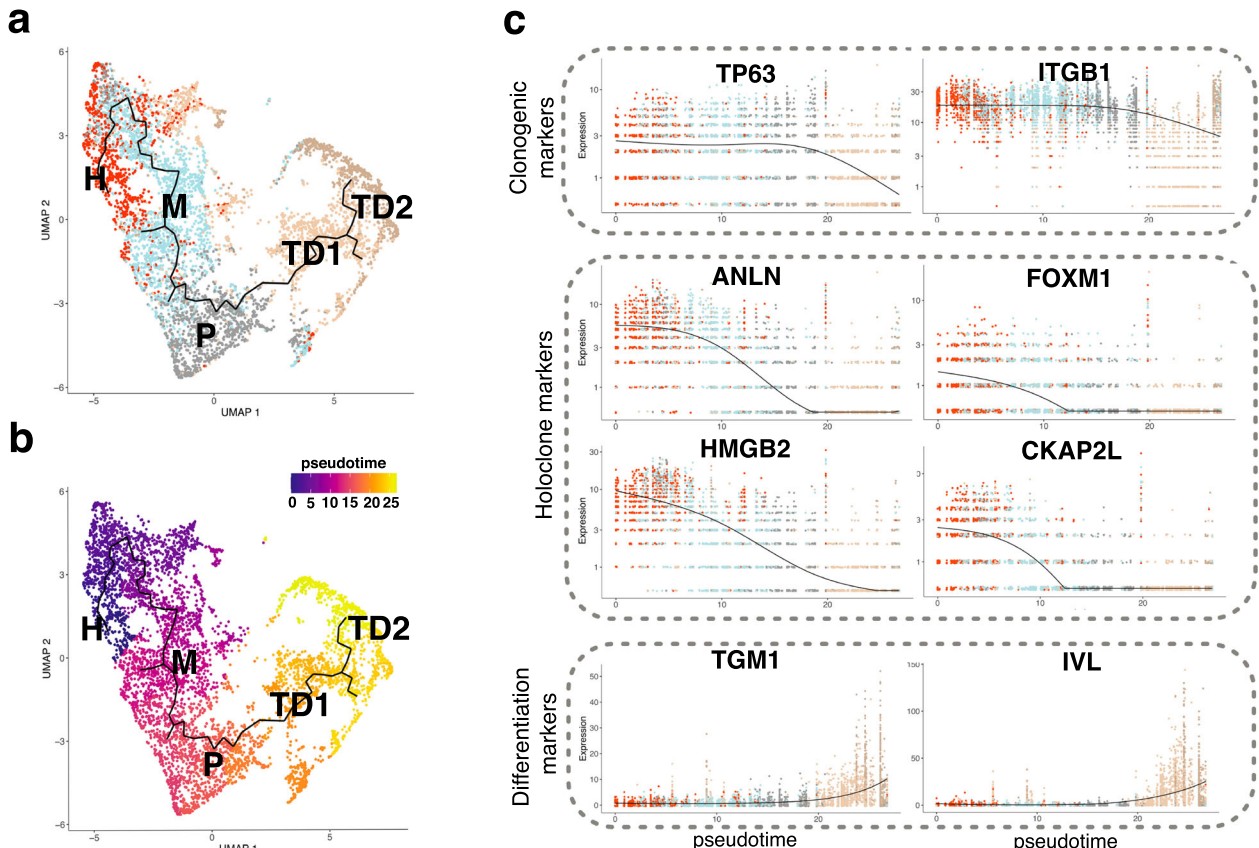

**Fig. 3 Single-cell RNA-seq analysis identifies a unique trajectory among the clusters. a** UMAP plot of cells annotated as keratinocytes identified at $t_1$ displaying the result of Monocle3 analysis. The identified trajectory is shown as a continuous black line. Cells are colored according to cluster identity determined with Seurat analysis. **b** Same as **a**, cells are colored according to pseudotime. **c** Kinetics plot showing expression values normalized and rounded by Monocle3 of two representative clonogenic markers (*TP63, ITGB1*), four representative markers present in the Holoclone signature (*ANLN, FOXM1, HMGB2, CKAP2L*) and two representative differentiation marker (*TGM1* and *IVL*) genes across pseudotime.

(K52) of holoclones. The relative amount of meroclones and paraclones was not significantly altered (Fig. 4c, left panel). Note that the initial population of holoclone-forming cells was much higher in K52 as compared to K5 and K71. Figure 4c (right panel) shows representative cultures generated by these transduced clones.

Three splice variants of FOXM1 have been described[24,25]. FOXM1-A is transcriptionally inactive[32]. FOXM1-B is the shortest isoform, is mainly expressed in cancer cells, and its expression is strictly dependent on cell cycle[33]. FOXM1-C, containing exon A1, is mainly expressed in normal cells ref. [32]. Indeed, primary human keratinocytes express almost exclusively FOXM1-C (Supplementary Fig. 4a, b) that is responsive to ERK inhibition obtained by 24 h exposure with U0126 (Supplementary Fig. 4c).

Primary keratinocytes were thus transduced with an empty vector (CNT) or a vectors carrying *FOXM1-B* and *FOXM1-C* under the control of a constitutive CMV promoter. Transgenic FOXM1 was expressed in the nucleus in virtually 100% of the cells (Fig. 4d). Of note, only enforced FOXM1-C (hereafter referred to as FOXM1) was able to induce the expression of p63 and survivin (Fig. 4e), without altering proliferation per se (analyzed through Edu/FxCycle incorporation, Supplementary Fig. 4d), growth rate, cell doublings, and number of cells at subconfluence during serial cultivation, all of which were measured during serial cultivation (Supplementary Fig. 4e). We then analyzed 253 clones isolated from 3 strains of transgenic FOXM1 cultures (K38, K49, and K57) during serial cultivation, passaging cells at confluence to hasten clonal conversion. After

3-5 passages, holoclone-forming cells were not detected in empty-vector-transduced cells (CNT), while all FOXM1-transduced keratinocyte cultures maintained a physiological percentage of holoclones (Fig. 4f, left panel). Figure 4f (right panel) shows representative cultures generated by these analyses.

Taken together, these data show that FOXM1 does not intrinsically affect keratinocyte proliferation, confirm that holoclones and meroclones have similar growth rate and behavior and points to a critical role of FOXM1 in upholding human epidermal stem cells, which represent a small percentage of all clonogenic keratinocytes.

We have recently shown that the YAP/TAZ signaling pathway sustains epidermal stem cells[16]. Enforced FOXM1 rescued the colony growth potential impaired by downregulation of YAP (see Methods, Fig. 5a and Supplementary Fig. 5a), prompting us to investigate whether FOXM1 is indeed regulated by YAP. Ablation of YAP by two independent siRNA (see Methods) inhibited transcription (Fig. 5b) and expression (Fig. 5c) of FOXM1. Such inhibition was evident already within 48 h after YAP-siRNA addition. YAP induces target genes through interaction with TEAD transcription factors. Disruption of YAP–TEAD interaction, attained by short-term exposure to Verteporfin, mimicked the effect of YAP ablation on FOXM1 transcription and expression (Fig. 5d). The effect of YAP in regulating FOXM1 expression seems to be direct, since both YAP and TEAD binds to the FOXM1 promoter on two TEAD recognition motifs (TRM), respectively, at −69 and −652 nucleotides from the transcription start site (TSS) (Fig. 5e)[34,35].

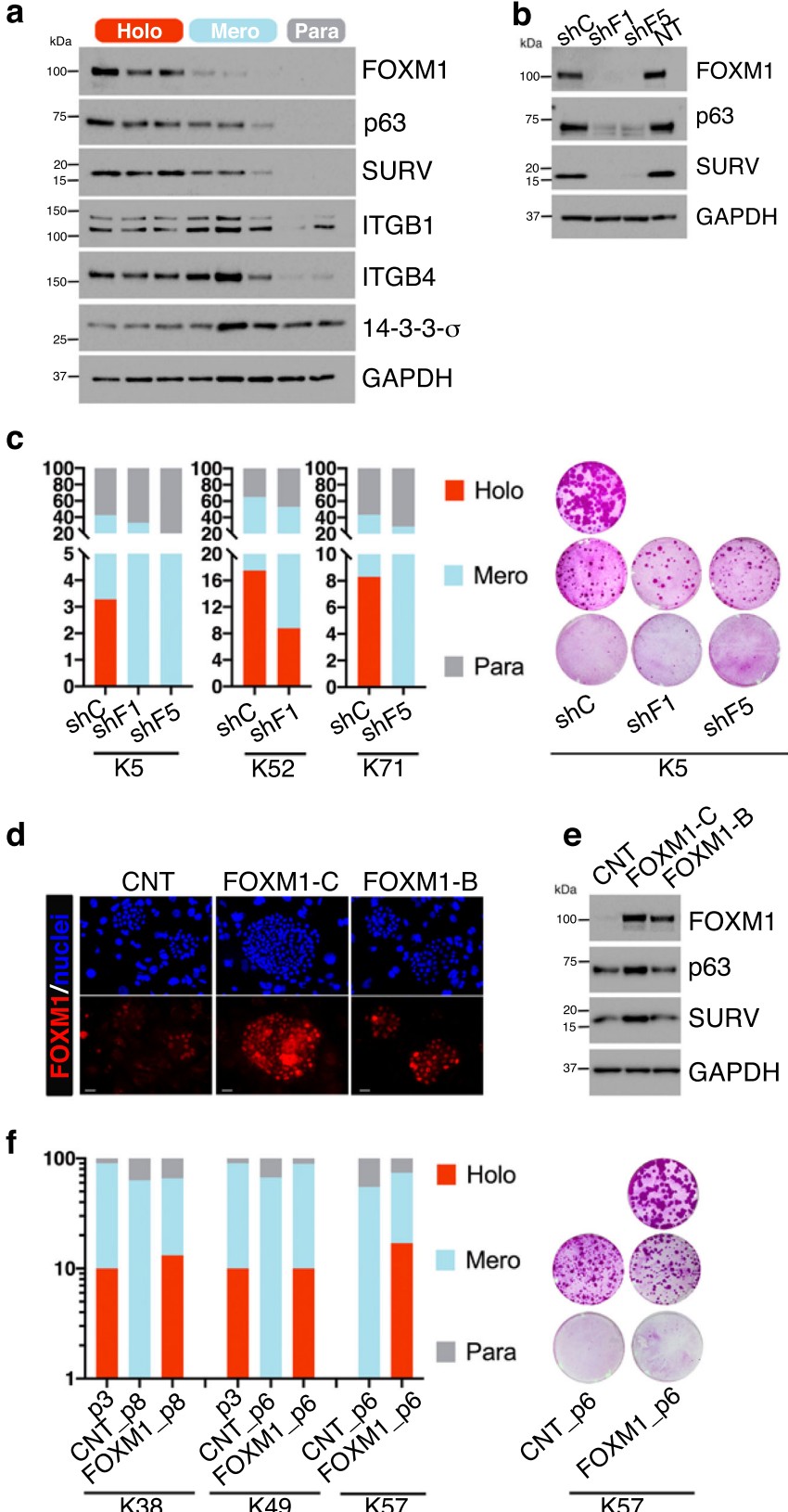

**FOXM1 sustains epidermal stem cells in JEB**. Generalized JEB is a devastating, often early lethal, genetic disorder characterized by structural and mechanical fragility of skin and mucosal membranes[36]. JEB is caused by mutations either in *LAMA3*, *LAMB3*, or *LAMC2* (which, together, encode laminin 332, also known as laminin 5), *ITGA6* and *ITGB4* (encoding α6β4 integrins), or *COL17A1* (encoding collagen XVII)[36]. The interaction of laminin 332 with α6β4 integrins at hemidesmosomes sustains epidermal stem cells through activation of the YAP/TAZ pathway; laminin 332-deficient JEB keratinocytes thus contain only phosphorylated, inactive YAP, which leads to epidermal stem cell depletion; enforced YAP recapitulates laminin 332-gene therapy in rescuing

**Fig. 4 FOXM1 underpins holoclone-forming cells. a** Western analysis of total cell extracts from cultures generated by holoclones, meroclones, and paraclones isolated by clonal analysis (see "Methods" and Fig. 1a) of sub-confluent normal human primary keratinocytes. Clonal conversion is marked by progressive decrease of FOXM1, p63, and survivin. β1 and β4 integrins remain stable in holoclones and meroclones and are downregulated in paraclones. 14-3-3σ progressively increases during clonal conversion (representative images of n = 3). **b** Western analysis of total cell extracts from keratinocytes transduced with the indicated shRNA shows that p63 and survivin are almost undetectable in FOXM1-depleted keratinocytes (representative images of n = 3). **c** Left: clonal analysis of shRNA-transduced clonogenic keratinocytes (see "Methods") from cultures initiated from three different biopsies of healthy skin (K5, K52, and K71). The percentage of holoclones, meroclones, and paraclones is indicated in red, light blue, and gray columns, respectively (n = 346 clones analyzed). Right: representative cultures generated by these transduced clones. **d** Representative image of immunofluorescence analysis of FOXM1 in 5-day colonies derived from control (CNT) and FOXM1-transduced keratinocytes. Scale bars, 20 μm. Representative images of three independent experiments are shown. **e** Western analysis of total cell extracts from cultures generated by control and FOXM1-C or B-transduced keratinocytes shows that p63 and survivin increase after FOXM1 overexpression (representative images of n = 3). **f** Left: clonal analysis of CNT and FOXM1-transduced clonogenic keratinocytes (see "Methods") performed at passage 3 (p3) and at passage 6 (p6) and 8 (p8) of cultures initiated from three different biopsies of healthy skin (K38, K49, and K57). The percentage of holoclones, meroclones, and paraclones is indicated in red, light blue, and gray columns, respectively (n = 253 clones analyzed). Right: representative cultures generated by these transduced clones.

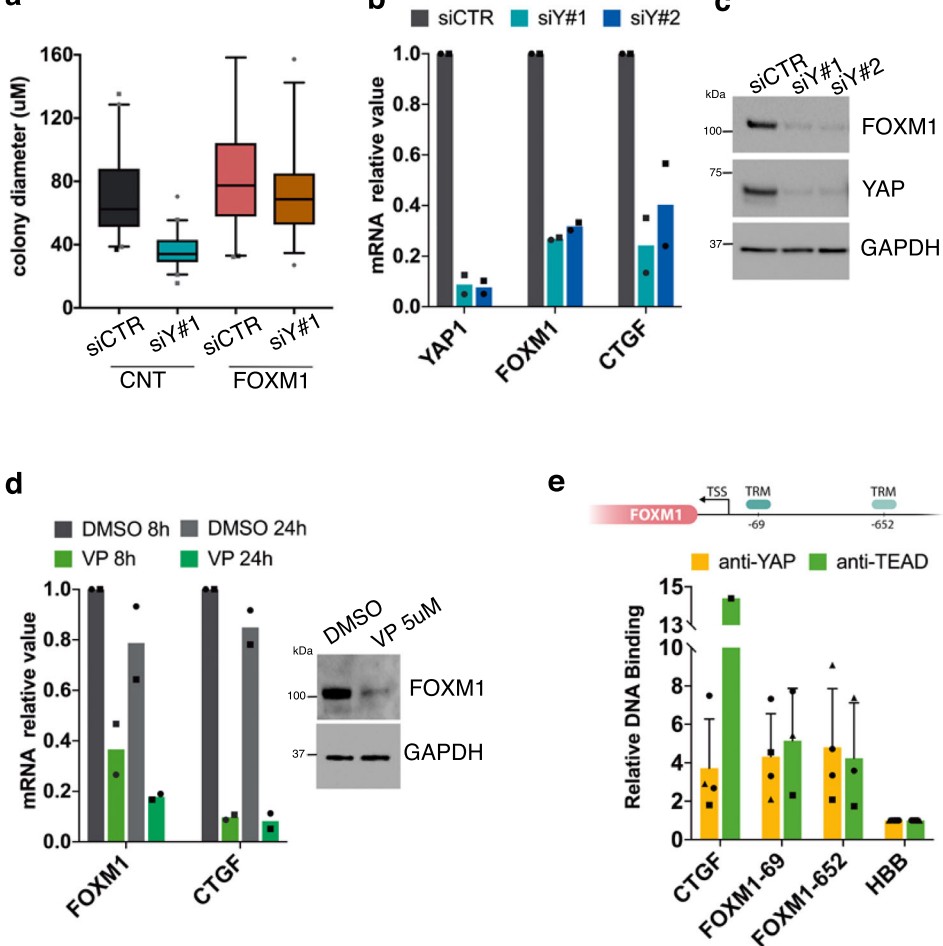

**Fig. 5 FOXM1 acts downstream of YAP. a** Box plot showing the diameter of colonies stained after 5 days of cultivation of CNT or FOXM1-transduced keratinocytes both transfected with control (siCTR) and YAP-specific (siY#1) siRNA. Data from one representative experiment. Enforced FOXM1 rescued the colony growth potential impaired by downregulation of YAP. Median and 5–95 percentile displayed. **b** qRT-PCR on mRNAs obtained from keratinocytes transfected with control (siCTR) and YAP-specific (siY#1, siY#2) siRNA. Expression levels of YAP1, FOXM1, and CTGF were normalized per GAPDH and given relative to siCTR (arbitrarily set to 1) (n = 2 biological replicates derived from independent human primary keratinocyte cultures, indicated with dots and squares, respectively). **c** Western analysis of total cell extracts from cultures treated with the indicated siRNA shows that FOXM1 expression is almost abolished after YAP depletion (representative images of n = 3). **d** left: qRT-PCR on mRNAs obtained from keratinocytes treated with Verteporfin (VP 5 μM) or vehicle (DMSO) for the indicated time. Expression levels of FOXM1 and CTGF were normalized per GAPDH and given relative to DMSO, 8 h (arbitrarily set to 1) (n = 2 biological replicates derived from independent human primary keratinocyte cultures, indicated with dots and squares, respectively). Right: Western analysis of total cell extracts from cultures treated with DMSO or VP 5 μM show that FOXM1 protein decreases after 24 h of treatment. **e** Up: Scheme showing two TEAD-recognition motifs (TRM), respectively, at −69 (ref. [35]) and −652 nucleotide[36] from transcription start site (TSS) into the FOXM1 promotorial region. Down: ChIP-qPCR showing YAP binding to the indicated sites in human primary keratinocytes. Relative DNA binding was calculated as a fraction of input and given relative to the negative region (HBB) arbitrarily set to 1; average and standard deviation displayed from four independent biological replicates, indicated with different shapes.

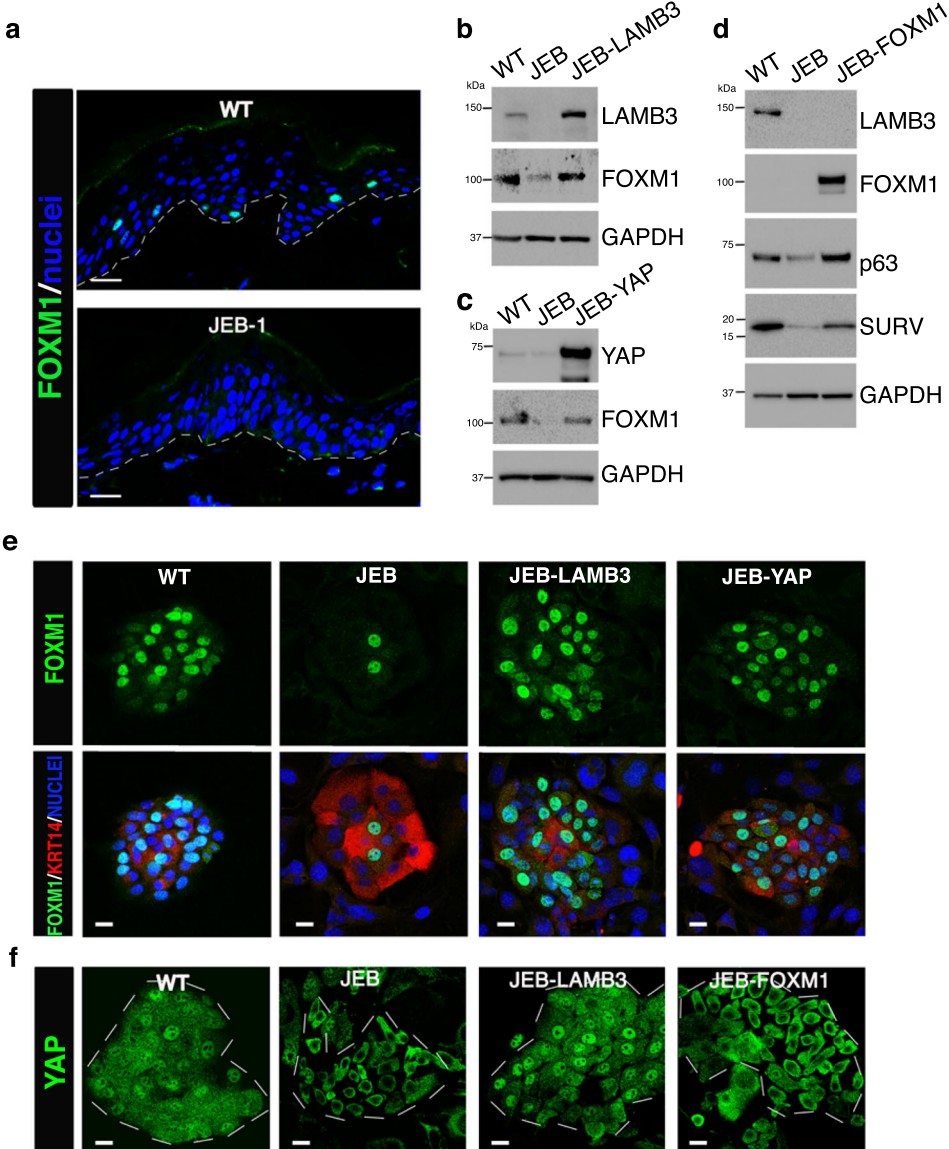

**Fig. 6 FOXM1 sustains epidermal stem cells in JEB. a** Immunofluorescence analysis of FOXM1 expression in 7-μm-thick skin sections prepared from normal skin (WT) and LAMB3-deficient JEB patient (JEB-1). DAPI (blue) stains nuclei. Dotted line marks the epidermal–dermal junction. Scale bars, 20 μm. Representative image of what observed in skin biopsy derived from five independent healthy donors and five sections randomly taken from JEB-1 skin biopsy. **b** Western analysis of total cell extracts from normal keratinocytes (WT), JEB, LAMB3-transduced, JEB cultures immunostained with indicated antibodies (representative images of n = 6). **c** Western analysis of total cell extracts from normal keratinocytes (WT), JEB, YAP-transduced, JEB cultures immunostained with indicated antibodies (representative images of n = 3). **d** Western analysis of total cell extracts from normal keratinocytes (WT), JEB, FOXM1-transduced, JEB cultures immunostained with indicated antibodies. Endogenous FOXM1 is not visible in WT extract due to the presence of high levels of enforced FOXM1 in FOXM1-transduced cells (representative images of n = 4). **e** Representative images of FOXM1 localization (green) and Pan-Keratin (red) in JEB, LAMB3-, and YAP-transduced JEB. DAPI (blue) stains nuclei. Scale bar 20 μm (representative images of n = 3). **f** Representative images of YAP localization (green) in normal (WT), JEB, LAMB3-, and FOXM1-transduced JEB. Colonies are indicated with white dotted lines. Scale bar 20 μm (representative images of n = 4).

JEB stem cells[16]. Of note, FOXM1 was virtually undetectable in skin sections prepared from JEB-1, a homozygous carrier of a c.1954delG mutation in the *LAMB3* gene (Fig. 6a).

*LAMB3*-dependent JEB is thus a suitable natural human model to substantiate the notion that FOXM1 sustains epidermal stem cells acting downstream of YAP. To this end, we used cells obtained from a JEB newborn (JEB), a double-heterozygous carrier of two mutations (c.2242G>T and c.823-1G>T) in the *LAMB3* gene. JEB was affected by a severe form of the disease, with poor prognosis. At birth, her epidermis expressed barely detectable laminin 332, but still contained a residual number of

stem cells[16]. JEB keratinocytes were transduced with gamma retroviral (γRV) expressing full-length *LAMB3* cDNA, or lentiviral vectors expressing either full-length *YAP* or *FOXM1*. Transduction efficiency of clonogenic cells was over 95% in all cases. FOXM1 expression was strongly decreased in JEB cells and rescued after *LAMB3*-gene therapy and enforced YAP, as observed by western blot (Fig. 6b, c) and immunofluorescence on isolated clones (Fig. 6e and Supplementary Fig. 5b). Strikingly, enforced FOXM1 was able to restore the expression of p63 and survivin (Fig. 6d) even in the absence of nuclear YAP, which is instead observed after *LAMB3*-mediated gene therapy (Fig. 6e).

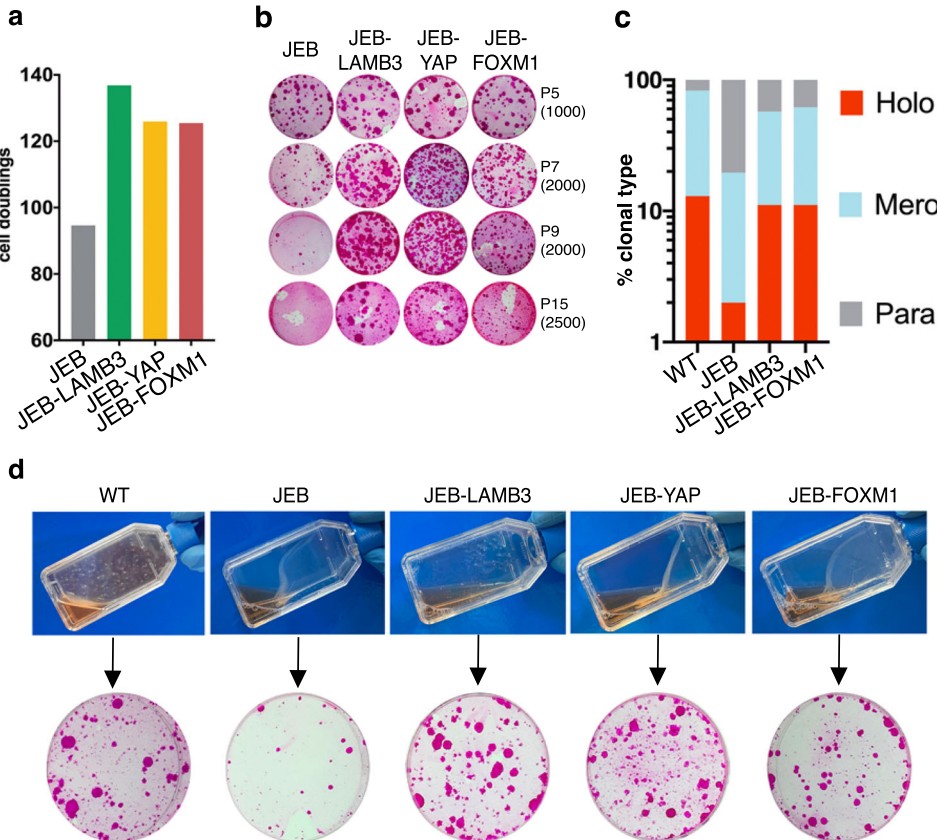

**Fig. 7 Uncoupling self-renewal from adhesion in FOXM1-overexpressing JEB derived keratinocytes. a** Calculation of cumulative cell doublings in JEB, LAMB3-corrected, YAP-transduced, FOXM1-transduced JEB cultures derived from one representative experiment. Number of cell doublings was calculated as described in "Methods". **b** CFE of keratinocyte cultures at passage 5, 7, 9, and 15 initiated from the control JEB patient and after LAMB3, YAP, or FOXM1 gene addiction. The number of cells per dish plated in the CFE is indicated between brackets and colonies were stained with Rhodamine B 12 days later. **c** Clonal analysis of normal keratinocytes (WT), untransduced, LAMB3-corrected, and FOXM1-transduced clonogenic JEB (see "Methods"). The percentage of holoclones, meroclones, and paraclones is indicated in red, light blue, and gray columns, respectively (*n* = 157 clones analyzed). **d** Top: adhesion of confluent cultured epidermal sheets prepared from normal keratinocytes (WT), JEB, LAMB3-corrected, YAP-transduced, FOXM1-transduced JEB29 cultures. The mere transportation of the flask from the incubator to the hood caused the spontaneous detachment of JEB and JEB-YAP and JEB-FOXM1 cultures, while WT and LAMB3-corrected keratinocytes remained firmly attached to the substrate. Bottom: CFE performed on the above cultures. The CFE was plated at a density of 4000 cells per dish, and colonies were stained with Rhodamine B 12 days later.

Due to the progressive loss of stem cells and the fast clonal conversion typical of JEB keratinocytes[16], JEB cells showed limited proliferative capacity (Fig. 7a) and a rapid decrease of their clonogenic ability (Fig. 7b), typically observed in clonally isolated meroclones and paraclones[6]. As expected, enforced YAP and *LAMB3*-gene therapy were equivalent in restoring both clonogenic ability and proliferative capacity of JEB epidermal cells (Fig. 7a, b).

Strikingly, enforced FOXM1 was able to recapitulate the effects of both *LAMB3*-gene therapy and enforced YAP, since it restored their clonogenic and proliferative abilities (Fig. 7a, b). Finally, enforced FOXM1 sustained JEB holoclone-forming cells at a comparable level to that of *LAMB3*-gene therapy (Fig. 7c).

FOXM1-mediated underpinning of epidermal stem cells occurred even with a disrupted epidermal adhesion machinery. Keratinocyte colonies eventually generate a cohesive epidermal sheet, which can be released from the vessel only after prolonged enzymatic treatment with the neutral proteases Dispase II[5,8,37–39]. In contrast, JEB epidermal sheets spontaneously detached from the culture vessel upon minimal shaking and lost their clonogenic ability (Fig. 7d, JEB). *LAMB3*-corrected JEB-cultured epidermal sheets remained firmly attached to the culture vessel and maintained their clonogenic ability (Fig. 7d, JEB-LAMB3). Strikingly, both enforced YAP and enforced FOXM1 did not restore the

adhesive properties of JEB epidermal sheets but fully preserved their clonogenic ability (Fig. 7d, JEB-YAP and JEB-FOXM1).

Taken together, these data confirm that FOXM1 acts downstream of YAP and can fully substitute for YAP in sustaining human epidermal stem cells to a similar extent of *LAMB3*-gene therapy. This holds true even after epidermal detachment, thus decoupling epidermal adhesion from stemness.

## Discussion

Autologous epidermal cultures have been used for over three decades to produce grafts that restore a functional epidermis on severely burned patients[10,11]. Notwithstanding appropriate clinical procedures, a critical evaluation of clinical successes and failures unveiled that the essential feature of the graft is the presence of an adequate number of stem cells. The long-term epithelial regeneration does not correlate with the total number of clonogenic cells, rather it depends on a defined number of holoclone-forming cells. In their absence, failures of epidermal regeneration are inevitable[11].

This said meroclones represent the vast majority of epidermal clonogenic cells and it has been virtually impossible to distinguish them from holoclones in vitro, since their morphology (both of colonies and cells within the colony), growth behavior, and rate of

proliferation are identical[4–7]. From a functional point of view, self-renewal is basically the only important feature distinguishing holoclones and meroclones. With the notable exception of limbal stem cells, which can be prospectively identified by ΔNp63α abundance[9,20,40], cumbersome, and expensive clonal analyses are still the only controls that can be used to detect stem cells in epidermal grafts.

Here we show that single-cell transcriptome analyses were indeed able to clearly distinguish epidermal holoclone-forming cells from other epidermal clonal types and cells entering the process of terminal differentiation. Holoclone-forming cells are enriched in genes found in embryonic stem cells and/or associated to cell cycle, chromosome segregation and stability, DNA repair, and, most notably, telomerase, which is selectively expressed by many somatic stem cells of tissues endowed with a constant turnover. Functional telomeres are essential features of long-lived stem cells, since they play fundamental roles in chromosome end protection and genome stability[41]. Holoclones were also enriched in genes involved in the organization of microtubules. Microtubule–actin interaction regulates cell shape and polarity in many cell types, including epithelial cells. Of note, microtubule organization impacts actin polymerization[42,43], which, in turn, plays a critical role in controlling epidermal stem cells. In fact, actin bundles are distributed radially in holoclones and circumferentially in paraclones. This actin filament dynamics are governed by Rac1 and are instrumental in controlling clonal conversion, hence stem cell maintenance[44].

Very recently, single cell RNA-seq has been performed on keratinocytes directly isolated from skin biopsies, hence on a resting, mature epidermis[45]. Instead, our profiling refers to clonogenic cultured keratinocytes mimicking wound healing and epidermal regeneration. Despite this fundamental difference, similar clusters of basal keratinocytes have been identified in both conditions: the BAS-II cluster is similar to our H cluster and the BAS-I cluster is similar to our M cluster[45]. The notion that the BAS-II cluster contains more cells than BAS-I is consistent with the notion that, as opposed to primary cultures, single clonogenic keratinocytes directly analyzed from a skin biopsy generate more holoclones than meroclones[46]. Similarly, and despite the notion that murine keratinocytes do not initiate the same clonal types identified in human cultures, both human holoclone-forming cells and the population of K14+ cells containing murine epidermal stem cells upregulate genes regulating DNA repair, cell cycle, chromosome segregation[3]. Our holoclone signature is enriched in the proliferative cluster of the murine stem cell/progenitor compartment[47,48]. Human H cluster (Fig. 3a, b) and murine proliferative/undifferentiated basal stem clusters[47] are marked by high expression of DIAPH3 and have been both identified as the starting point of the differentiation trajectory passing thought the TA progenitors and then differentiated cells.

Computational analysis of scRNA-seq data identified a continuous trajectory showing that holoclone-forming cells generate meroclone- and paraclone-forming cells (eventually producing terminally differentiated cells), strengthening the notion that long-lived self-renewing stem cells generate pools of TA progenitors[5], which are known to play a role in epidermal regeneration during wound-healing processes. Indeed, meroclones and paraclones were enriched in sets of genes related to wound healing, corroborating the notion that clonogenic keratinocytes derived directly from a biopsy taken from normal, unwounded skin generate mostly holoclones, whereas keratinocytes cloned from wounded skin generate predominantly meroclones and paraclones[46].

Bulk and single-cell transcriptional profiling allowed to identify FOXM1 as a key transcription factor regulating holoclone-forming cells. Indeed, (i) FOXM1 is expressed virtually only in holoclones, (ii) its ablation causes selective disappearance of holoclone-forming cells, and (iii) enforced FOXM1 hampers clonal conversion, hence sustaining holoclone-forming cells overtime.

FOXM1 has been studied mainly in cancer cells, including transformed human keratinocytes, where it drives G2/M transition. But an important role of FOXM1 has been also reported in many stem cells[49]. In fact, FOXM1 (i) regulates the G2/M phase of the mammalian embryonic stem cell cycle and plays a role in protecting such cells from oxidative stress[26]; (ii) controls a miRNA network involved in the self-renewal of murine neural stem cells[28]; (iii) regulates Nurr1-mediated self-renewal of murine hematopoietic stem cells[29]; (iv) controls satellite cell-mediated murine muscle regeneration through its interaction with the Wnt/β-catenin pathway[30]; (v) is required for the proliferation and differentiation of murine Clara cells as well as for proper differentiation of airways epithelial, hence for long-term maintenance of the bronchiolar epithelium[31]; (vi) is among the transcription factors marking highly proliferative murine clonogenic keratinocytes[50]. In cancer cells, FOXM1 regulates mostly cell proliferation.

In contrast, on human primary clonogenic keratinocytes, FOXM1 seems to have little effect on proliferation per se, since short-term ablation of FOXM1 did not affect the number of cells nor the number of mitotic figures. Even long-term ablation of the protein had virtually no effect on both clonogenic ability and proliferation of TA progenitors, which generated colonies of the expected size. Strikingly, FOXM1 depletion selectively and rapidly impaired mainly holoclones. FOXM1 (and YAP) is thus instrumental to the clonal conversion process, which regulate the long-term proliferative potential and the self-renewal, all of which mark specifically epidermal stem cells[5,23].

We show that FOXM1 executes its control on holoclone-forming cells downstream of YAP, confirming the role of YAP in sustaining epidermal stem cells and identifying FOXM1 as a YAP target, at least in human epidermis. These findings are in agreement with previous reports showing that YAP binds FOXM1 promotorial regions through TEAD in several tumors, as sarcoma[34], mesothelioma[35], colorectal cancer[51], and liver cancer[52].

Of note, the YAP dysregulation-dependent loss of epidermal stem cells specifically marking JEB[16] can be ascribed to the loss of expression of FOXM1. Indeed, enforced FOXM1 recapitulates LAMB3-mediated gene therapy of JEB and can substitute for YAP in preserving JEB epidermal stem cells, even in the absence of cell adhesion. Thus, these data add a new element to the YAP/TAZ-dependent regulation of human epidermal stem cells.

Intriguingly, however, FOXM1 does not seem to fully substitute for YAP. Indeed, enforced YAP—or ablation of 14-3-3σ, which blocks YAP sequestration in the cytoplasm—induce complete full stop of clonal conversion and indefinite keratinocyte proliferation[16,22]. Instead, enforced FOXM1 do not induce bypass of replicative senescence but only prolonged keratinocyte proliferative potential (see Fig. 7b), hence suggesting that clonal conversion was greatly slackened but not fully halted and that FOXM1 is responsible for some, but not all, effects of the YAP/TAZ pathway on epidermal clonal dynamics.

The possibility of prospectively distinguishing holoclone-forming cells from other epidermal clonal types and the identification of proteins specifically expressed by the different types of clonogenic keratinocytes will allow a better control of the stem cell content on cultured epidermal grafts and drive a better development of combined ex vivo cell and gene therapy for different forms of epidermolysis bullosa and other genetic skin diseases.

## Methods

**Human tissues**. All human tissues were collected after informed consent for use of tissues in research and in compliance with Italian regulations (Comitato Etico dell'Area Vasta Emilia Nord, number 178/09 for healthy donor skin samples and number 124/2016 skin biopsies obtained from patients affected by JEB).

**Primary human cell cultures from healthy donors**. Human skin samples from surgical waste (abdominoplasty or mammoplasty) were collected and anonymized. Briefly, skin biopsies were minced and treated with 0.05% trypsin/0.01%EDTA for 4 h at 37 °C. Every 30 min keratiocytes were collected, plated ($2.5$–$3 \times 10^4$/cm$^2$) on lethally irradiated 3T3-J2 cells ($2.4 \times 10^4$/cm$^2$), and grown at 37 °C, 5% CO$_2$ in humidified atmosphere in Dulbecco's modified Eagle's (DMEM) and Ham's F12 media (3:1 mixture) containing fetal bovine serum (FBS) (10%), penicillin–streptomycin (50 lU/ml), glutamine (4 mM), adenine (0.18 mM), insulin (5 mg/ml), cholera toxin (0.1 nM), hydrocortisone (0.4 mg/ml), triiodothyronine (Liothyronine Sodium) (2 nM), epidermal growth factor (EGF, 10 ng/ml) (KGM)[5,16]. When subconfluent, cell cultures were serially propagated until senescence»[16].

**Primary JEB cultures**. A skin biopsy (1 cm$^2$) has been collected from a LAMB3-dependent JEB patient (1-month-old) and cultivated as described above.

**3T3-J2 cell line**. Mouse 3T3-J2 cells were a gift from Prof. Howard Green, Harvard Medical School (Boston, MA, USA). Fibroblasts were cultivated in DMEM supplemented with 10% gamma-irradiated donor adult bovine serum, penicillin–streptomycin (50 IU/ml) and glutamine (4 mM). EUFETS, GmbH, (Idar-Oberstein, Germany) produced a GMP clinical grade 3T3-J2 cell bank. That have been authorized for clinical use by national and European regulatory authorities.

**MFG-LAMB3-packaging cell line**. Full-length 3.6-kb LAMB3 cDNA (Gene Bank Accession #Q13751, fully sequenced) were cloned under the control of the MLV LTR into MFG-backbone[63]. The amphotropic Gp+envAm12 packaging cell line was used to generate the Am12-MGFLAMB3 producer cell line[23]. Accordingly with IHC guidelines, Molmed S.p.A, Milan, Italy produced a GMP-grad master cell bank of a high-titer packaging clone (GP+envAm12-LAMB3 cells), The packaging cells ere cultured in DMEM supplemented with10% irradiated fetal bovine serum, penicillin/streptomycin (50 IU/ml), glutamine (2 mM), and subjected to quality and safety tests under GMP conditions.

**Generation of genetically corrected epidermal keratinocytes**. Sub-confluent JEB mass cultures were treated with 0.05% trypsin and 0.01% EDTA for 15–20 min at 37 °C and keratinocytes ($1.33 \times 10^4$ cells/cm$^2$ were co-cultured in KGM with $8.3 \times 10^4$ cells/cm$^2$ lethally irradiated fibroblasts (a mixture of 3T3-J2 cells and GP +envAm12-LAMB3 in 1:2 ratio). Three days later, cells were trypsinized and cultured on a 3T3-J2 feeder layer in KGM. At each step, colony-forming efficiency was determined by plating 1000 cells into an indicator dish stained with rhodamine B after 12 days.

**Clonal analysis**. Sub-confluent keratinocytes mass cultures were trypsinized and 0.5–1 cells was plated into each well of a 96-well plate after serial dilution. Single clones were cultivated for 7 days and treated with 0.05% trypsin and 0.01% EDTA at 37 °C for 15–20 min. One-quarter of the clone was plated into an indicator dish, cultivated for 12 days and stained with rhodamine B to classify the clonal type.

The remaining three-quarters was sub-cultivated into an adequate plastic support and used for further analyses, as microarray or protein/RNA extraction with NucleoSpin RNA/protein kit from Macherey-Nagel.

**Colony-forming efficiency, population doublings, growth rate, and clone size**. Colony-forming efficiency was calculated at each passage from the indicator dish stained with rhodamine B after 12 days of cultivation. Percentage of clonogenic cells correspond to the number of colonies on the total amount of plated cells; percentage of abortive colonies was calculated as a number of abortive colonies on the total number of colonies. The following formula was used to score the number of cell doublings: $x = 3.322 \log N/N_o$, where $N$ is the total number of keratinocytes collected at each passage and $N_o$ is the number of clonogenic cells plated[7]. CFE was determined during serial cultivation at each passage. Growth rates were calculated using the following formula log ((number of collected keratinocytes)/(number of seeded keratinocytes))/log(2)[53]. Clone size has been calculated using ×10 objective and ZEN plug-in

**Lentiviral production and primary human keratinocyte infection**. HEK293T cells were cultivated in DMEM supplemented with 10% FBS, 1% Pen/ Strep, 1% glutamine (Thermo Fisher). HEK293T cells were transiently transfected with pMD2-VSVG, pPAX2, and the lentiviral plasmid by using calcium phosphate transfection. Lentiviral particles were collected after 48 h post-transfection, filtered through 0.45-µm-pore cellulose acetate filters, and concentrated by ultra-centrifugation. Primary human keratinocytes were transduced with 8 mg/ml

polybrene (Sigma) at MOI10. Keratinocytes were passaged at subconfluence and collected for further analyses.

**Microarray**. Quality and quantity of the 60 nucleic acid samples were established using the Agilent Bioanalyzer RNA Nano kit and the Nanodrop 1000, respectively. GeneChip 3′ IVT PLUS Reagent Kit (Thermo Scientific) provided sufficient amplified double-stranded cDNA for each holoclone, meroclone, or paraclone total RNA sample, letting to in vitro transcribe it to labeled cRNA, that could be fragmented and therefore hybridized onto single GeneChip™ Human Genome U133 Plus 2.0 Array (ThermoScientific), following the manufacturer's indications. To avoid batch effect among samples, i.e. no technical sources of variation added to the samples during handling, samples were randomized during RNA isolation, sample preparation, and hybridization/scanning working sessions. All samples were processed with the same reagents lot number, when available. Sample and hybridization quality controls were carried out with Transcriptome Analysis Console (TAC, ThermoScientific) to verify complete and unbiased coverage of the transcriptome.

**Gene expression analysis**. Microarray analyses were performed in R (version 3.5.0) using Bioconductor libraries and R statistical packages. Probe level signals were converted to expression values using the robust multi-array average procedure RMA[54] comprised in the Bioconductor *affy* package and a custom chip definition file based on the Entrez gene database (Brainarray version 23[55]). PCA has been performed using the function *prcomp* of R *stats* package. Global unsupervised clustering was performed using the function *hclust* of R *stats* package with Pearson correlation as distance metric and average agglomeration method. Before unsupervised PCA and clustering, to reduce the effect of noise from non-varying genes, we retained the top 10% variable genes, removing those genes with a coefficient of variation smaller than the 90th percentile of the coefficients of variation in the entire dataset. Differentially expressed genes were identified using the Significance Analysis of Microarray algorithm coded in the *samr* R package[56]. In SAM, we estimated the percentage of false-positive predictions (i.e., FDR) with 1000 permutations and identified as differentially expressed those genes with FDR ≤ 5% and absolute fold change larger than a selected threshold (e.g. ≥1.5) in the comparison of holoclones vs meroclones and holoclones vs paraclones. To identify expression hallmarks of the holoclones, we defined as the holoclone signature those 526 genes upregulated in both comparisons (Supplementary Data 1).

Gene Ontology (GO) analyses for Biological Process category were performed using DAVID[57]. GO terms with a Benjamini–Hochberg corrected *p* value ≤5% were considered significantly enriched. The fold enrichment is defined as the ratio of the proportions of genes associated with a GO term in the gene list and in the background, respectively. Functional over-representation analysis was performed using Gene Set Enrichment Analysis (GSEA; http://software.broadinstitute.org/gsea/index.jsp[58]), the curated gene sets of the Molecular Signatures Database (MSigDB version 7.0[59]), derived from the Pathway Interaction Database (PID) collection, and a set of previously published signatures comprising gene sets of stem cells and for the activation of signal transduction pathways[60]. The complete list of gene sets used in this study is provided in Supplementary Data 2. Gene sets were considered significantly enriched at FDR ≤ 5% when using *Signal2Noise* as metric and 1000 permutations of phenotypes. The dot plot, showing the most significantly enriched gene sets in Holoclones, was generated using the *ggplot* function of the *ggplot2* R package.

**Encapsulation with 10X Genomics chromium system and single-cell RT**. The primary culture used for clonal analysis was also used for single-cell RNA-seq assay. Keratinocytes were detached with trypsin for 15–20 min in order to obtain a single-cell suspension and pelleted in culture medium. Cells were then suspended in 1× phosphate-buffered saline (PBS) with 0.04% BSA and filtered with 70 µM cell filter in order to discard any clamp or cell cluster. Cell suspension were then visualized and counted with trypan blue using a Countess™ II Automated Cell Counter to get a precise estimation of total number of cells and of cells concentration. Afterwards we loaded about 10,000 cells of each sample into one channel of the Chromium Chip B using the Single Cell reagent kit v3 (10X Genomic) for Gel bead Emulsion generation into the Chromium system. Following capture and lysis, cDNA was synthesized and amplified for 14 cycles following the manufacturer's protocol. Fifty nanograms of the amplified cDNA were then used for each sample to construct Illumina sequencing libraries. Sequencing was performed on the NextSeq550 Illumina sequencing platform following the 10X Genomics instruction for read generation, reaching at least 50,000 reads as mean reads per cell.

**Bioinformatic analysis on single-cell RNA seq data**. The Cell Ranger Count pipeline (version 3.1.0) was used to align reads of the dataset at $t_1$ to the reference transcriptome (GRCh38) and to calculate UMI counts from the mapped reads. Expression data were imported in R and analyzed using Seurat (version 3.1.5[61]) R package. Low quality and multiplet cells were identified as outliers within the distribution of the number of genes, UMI counts, and percent of reads mapping on mitochondrial genes per cell, and subsequently discarded. Samples were integrated using the Seurat version 3.1.5 integration strategy. Prior to dimensional reduction with PCA, a cell cycle score was assigned to each cell and regressed out. We

selected 16 principal components for cluster analysis and visualization with uniform manifold approximation and projection (UMAP). Clusters representing fibroblasts were identified using the expression level of vimentin. We discarded two small keratinocyte clusters representing low-quality cells and cells with high levels of stress-response-related genes. We classified the remaining five keratinocyte clusters into holoclones, meroclones, paraclones, and terminally differentiated cells monitoring the expression of known markers and the expression of the holoclone signature, calculated as the average expression of its constituent 526 genes. Functional over-representation analysis was performed using GSEA in preranked mode and the same gene sets used for microarray expression data. Genes were ranked according to the log fold change in the comparison of the holoclone cluster and each of the other two clonogenic clusters.

Trajectory analysis was performed on these clusters using Monocle3 R package[62]. Kinetics plots were generated with the Monocle3 *plot_genes_in_pseudotime* function. Single-cell transcriptomic data of serially cultured $t_2$ keratinocytes were processed and integrated using the same procedure described above. Cells were classified using the annotated dataset at $t_1$ as reference and the *FindTransferAnchors* and *TransferData* functions in Seurat with default parameters. Expression data are available in Gene Expression Omnibus with accession number GSE155817.

**Western blotting**. Feeder layer was removed in 20 mM cold PBS/EDTA. Keratinocytes were collected by scraping in 1× RIPA buffer (Sigma Aldrich) supplemented with phosphatase and Protease Inhibitor Cocktail (Thermo Fisher). BCA kits (Pierce) were used to quantify the total protein amount. The same number of proteins was loaded in 4–12% NuPAGE Bis-Tris Gels or 10% NuPage Tris-Acetate Gels (Thermo Fisher) and transferred 100 V at 4 °C for 2 h onto nitrocellulose membrane (Millipore). Membranes were treated with blocking solution (5% (w/v) non-fat milk in 0.01% (v/v) Tween-20 in PBS 1×). Primary antibodies were diluted in blocking solution as indicated in Supplementary Table 1 and added overnight at 4 °C to the membranes. Secondary antibodies were diluted in Blocking solution as indicated in Supplementary Table 1 and added to the corresponding membranes for 1 h at room temperature. Signal was visualized with Clarity Western ECL substrate (Bio-Rad) using ChemiDoc (Bio-Rad) and ImagLabs software. Gray background on the images was homogeneously added for graphical purpose. Uncropped blots can be found in the Source data file.

**Immunofluorescence**. For immunofluorescence analysis, keratinocytes were plated at 1000/wells onto glass coverslips and grown as previously described. Cells were fixed with PFA 3% for 10 min at room temperature, carefully washed with 1× PBS and permeabilized in PBS/Triton 0.5% for 15 min. Blocking solution (FBS 5% BSA 2% in PBS/Triton 0.1%) was added for 1 h at r.t. Primary antibodies were diluted in Blocking solution as described in Supplementary Table 1 and added to the samples overnight at 4 °C. Secondary antibodies were diluted in Blocking solution as described in Supplementary Table 1 and added to the samples for 1 h at r.t. Cell nuclei were stained with DAPI. Dako Mounting medium was used to mount coverslips. Zeiss confocal microscope LSM510meta with a Zeiss EC Plan-Neofluar ×40/1.3 oil immersion objective was used to visualize fluorescent signals.

For in vivo immunofluorescence, skin biopsies derived from healthy donors or JEB-1 were washed in PBS, embedded in Killik-OCT cryostat embedding medium (Bio-Optica), and frozen. Immunofluorescence was performed using Leica Bond RX on 7-μm skin sections. In brief sections were fixed in PFA 3%, permeabilized with HIER 2 solution (Buffer citrate, EDTA pH 9) for 15 min at 80 °C, and blocked with PBS–BSA 2% for 30 min at 37 °C with Blocking solution (PBS–BSA 2%). Primary antibody, diluted in Blocking solution, was added to skin sections for 1 h at 37 °C. Sections were washed three times in Bond Wash solution (commercial TBS made for Bond RX by Leica) and incubated with Alexa Fluor 488 donkey anti-rabbit (Thermo Fisher) diluted 1:2.000 in Blocking solution for 30 min at 37 °C. Cell nuclei were stained with DAPI for 3 min at room temperatures. Glasses were then mounted with Dako Mounting medium and fluorescent signals were monitored under a Zeiss confocal microscope LSM510meta with a Zeiss EC Plan-Neofluar ×40 3/1.3 oil immersion objective. The antibodies used for immunofluorescence are described in Supplementary Table 1.

**Flow cytometry biparametric staining for cell cycle analysis**. Cell cycle analysis was performed using Click-iT™ EdU Flow Cytometry Assay Kit by Invitrogen according to the manufacturer's protocol. In brief, EdU was added to the cells as per dilution 1:1000, after 2 h cells were harvested and stained for 3t3-feeder cells. Cells were then fixed, permeabilized, incubated with Click-iT reaction cocktail and, at last, stained with FxCycleTMViolet. Stained cells were analyzed with BD FACSCanto II, BD FACSDiva Software v6,1,3 and FlowJo v10.

**Plasmid constructs**. For FOXM1 overexpression, cDNA of FOXM1-A, -B and -C isoforms were cloned in pCDH1 expression plasmid under the control of a constitutive CMV promoter (gift from Weiguo Hu). Empty backbone was used as control. Inducible YAP constructs (YAP1-WT) were derived from De Rosa et al. (2019)[16].

**Transient transfection**. For FOXM1 and YAP1 a total amount of 100 nM of specific siRNA (Silencer Select, Thermo Fisher; Supplementary Table 2) were transfected by Lipofectamine RNAiMAX (Thermo Fisher) for 5 h in the absence of serum. After 5 h the medium was changed and replaced in Kc medium. The cells were collected at the indicated time after transfection in RIPA buffer (Sigma) for protein extract or in Lysis buffer (Thermo Fisher) for mRNA collection.

**RNA extraction and real-time qPCR**. RNA for microarray analysis were extracted with NucleoSpin RNA/protein kit from Macherey-Nagel. For real-time qPCR, total RNA was isolated from cultured cells using the PureLink RNA Mini Kit (Thermo Fisher). Complementary DNA was generated using the SuperScript VILO cDNA Synthesis Kit (Thermo Fisher). Real-time qPCR analyses were carried out on triplicate samplings of retrotranscribed cDNAs with Taqman Universal PCR Master mix or PowerUP SYBR green master mix (Thermo Fisher) on 7900H Real-Time PCR System (Thermo Fisher). Expression levels are given relative to GAPDH. List of TaqMan probes (Thermo Fisher) and oligonucleotide (custom made by Eurofins genomics) is provided in Supplementary Tables 3 and 4, respectively. Data were analyzed with RQ Manager Software 1.2.2 and visualized with Prism 7.

**Chromatin immunoprecipitation**. For chromatin immunoprecipitation ChIP-kit from Abcam (ab500) was used following the manufacturer's instruction. Briefly cells were crosslinked with 1% formaldehyde (Sigma) in culture medium for 10 min at room temperature, and chromatin from lysed nuclei was sheared to 200–600 bp fragments using a Branson Sonifier. Chromatin derived from $4 \times 10^6$ cells was incubated with indicated antibodies (Supplementary Table 1) overnight at 4 °C. Antibody/antigen complexes were recovered with ProteinA/G beads for 2 h at 4 °C. Quantitative real-time PCR was carried out on a 7900H thermal cycler with custom-made oligonucleotides with PowerUP SYBR green master mix (Thermo Fisher); each sample was analyzed in triplicate. The amount of immunoprecipitated DNA in each sample was determined as the fraction of the input (amplification efficiency (Ct INPUT_Ct ChIP)). Primers are listed in Supplementary Table 4.

**Reporting summary**. Further information on research design is available in the Nature Research Reporting Summary linked to this article.

## Data availability

Microarray and scRNA-seq data have been deposited in the Gene Expression Omnibus database under accession code: GSE155817. Database used: Molecular Signature Database: https://www.gsea-msigdb.org/gsea/index.jsp and Brainarray: http://brainarray.mbni.med.umich.edu/Brainarray/Database/CustomCDF/23.0.0/entrezg.asp. Source data are provided with this paper.

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

## Acknowledgements

This work was supported by the Italian Ministry of Education, University and Research (MIUR) (CTN01_E48C13000140008), POR-FESR 2014-2020 Regione Emilia-Romagna (E8IJ10000120007 and E92I16000220005), Lombardia è Ricerca Award, Debra Alto Adige to M.D.L. and FAR 2019 (E54I19002000001) to E.E.. We thank Giorgio De Santis for providing us with healthy skin biopsies, Chiara Fiorentini and Giovanni Pellacani for JEB biopsies and their clinical work on JEB patients. We thank Ivanna Nesteruk, Giorgia Rizzardi, Michele Palamenghi, Michael Giovanardi, and Antonio Di Rocco for their help during the project. We thank Weiguo Hu, Stefano Piccolo, and Sirio Dupont for sharing plasmids.

## Author contributions

E.E. defined strategic procedures, performed experiments, analyzed data, assembled all input data, and edited the manuscript. A.S.S. performed experiments and clonal analyses and analyzed data. R.C., S.C., M.P.P., and I.S. performed experiments. E. Tenedini and E. Tagliafico performed microarray experiments and conduct preliminary bioinformatic analysis. C.P. performed single-seq RNA seq experiments. M.F. and S. Bicciato conducted all bioinformatics analyses. S. Bondanza performed clonal analyses. M.D.L. coordinated the study, defined strategic procedures, administered the experiments, and wrote the manuscript.

## Competing interests

M.D.L. is co-founder and member of the Board of Directors of Holostem Terapie Avanzate (HTA), s.r.l, Modena, Italy, as well as consultants for J-TEC-Japan Tissue Engineering, Ltd. The remaining authors declare no competing interests.
