## [Peer Review File · Nature Communications]

Reviewers' Comments:

Reviewer #1:

Remarks to the Author:

This is very interesting manuscript, where the authors have shown that FOXM1 acts as regulator of human epidermal stem cells by using different approaches including clonogenic assays, genome wide transcriptomic analysis and single cell RNA sequencing, functional assays using gain and loss of function. The authors demonstrated the role of FOXM1 and its regulation by YAP in sustaining human epidermal stem cell fate. This is a very thorough and well executed study that provide important insights into the molecular mechanisms regulating human epidermal stem cells. The study is well controlled and very convincing. I recommend publication of this study to nature communications.

Comments

It will be interesting to compare the molecular features of holoclones based on bulk and single cell transcriptomic with the various transcriptomic data (populations and single cell RNA-seq) that have been published in mouse epidermal stem cells to define common epidermal stem cell and progenitor markers across species. There are various studies that have provided such bulk and single cell RNA-seq of mouse epidermal stem cells and progenitors during homeostasis, development and tissue repair (Mascre et al., 2012, Sánchez-Danés et al., Nature 2016, Joost Cell Systems 2016, Joost Cell Reports 2018, Dekoninck et al., Cell 2020, Aragona et al., Nature2020). It will be interesting to correlate holoclone and paraclone specific markers of human epidermal stem cells with mouse stem and progenitor cells.

Reviewer #2:

Remarks to the Author:

The manuscript has a refreshing experimental set-up and the strategy to discriminate the different stem cell pools prior to terminally differentiated keratinocytes is well-designed. A potential new signature for holoclones would be welcome, as to be better able to serve those in need of skin transplants.

My main concerns therefore do not lie in the potential impact, nor in the execution of the study. Instead, they much more regard the true meaning and value of the FOXM1-G2/M signature in being unique to, and thus useful to detect, holoclones vs. paraclones and + terminally differentiated keratinocytes (and to a lesser extent meroclones). Holoclones divide. More YAP1 means more FOXM1 and therefore more division. But this still only confirms FOXM1 is a cell cycle gene; not that FOXM1 is unique to holoclones. Promoting proliferation of holoclones would of course be of great value, but this per sé would then be the main difference between holoclones and the other groups: G2/M status.

More specifically:

1. FOXM1 is a well-known G2/M protein. It regulates G2/M transition, amongst others, through regulation of AuroraB kinase and cyclins as CCNA (see fig. 1+2 as target genes also picked up here). The fact that all of these are elevated in the holoclone signature would be in line with the fact that it is the holoclone cells that are still proliferating and are going through G2/M phase of the cell cycle. A BrdU/EdU staining would then also likely be able to discriminate holoclones from paraclones (and to a lesser extent meroclones). To rule out we are simply looking at proliferation (G/M status), a BrdU/EdU incorporation could easily be performed, and/or a cell cycle distribution assay by cytometry.

I suspect the authors are looking at proliferation (G2/M) markers, as the holoclones are able to divide. But that signature is not unique to holoclones (as all dividing cells will have it) and it would then also not be a specific gene signature that can be used to discriminate these cells in a complex

mixture in vivo/ex vivo. In other words, can this signature be used to detect holoclones in vivo?
E.g. after injury?

2. In line with this. I suspect it is the culture media that dictates the holoclones to develop the FOXM1-G2/M signature. Would this signature still remain in holoclones that are placed on low growth factor media?

3. FOXM1b and c differ in ERK phosphorylation sites – which are cell cycle controlled. Of note. Also FOXM1c is expressed in cancer cells, e.g melanoma.

Does FOXM1b expression give the same phenotype in Fig. 4? This would also to an extent address the concern of FOXM1 here being merely a proliferation factor.

4. Stable expression of FOXM1 would increase the change of cancer. This would impair the use of ectopic FOXM1 (and LAMB3/YAP1) expression to counteract JEB. However, there are transient ways of enforcing these proteins. Though arguably not the best when wanting to use these clones for transplantation purposes, proof-of-concept could for instance be obtained with Nicotine, which has been claimed to promote FOXM1 expression in keratinocytes. Activators of YAP/TAZ can also be explored to study the importance of FOXM1 in holoclone definition.

5. JEB-LAMB3 seems to provide much larger/stable colonies than JEB-FOXM1. Would it not be better for the authors to investigate transient LAMB3 expression in promoting holoclone abundance?

6. In line with this, in fig. 7 it was not shown whether FOXM1 was still elevated in the JEB-YAP and dJEB-LAMB3 clones, making it a bit hard to conclude this is all caused by FOXM1, instead of other actions from LAMB3

Textual / individual results:

1. Why do the authors group telomerase with the DNA-repair genes?

2. It would be good to include a concluding remark in the abstract on the translational potential of the findings of FOXM1 expression for JEB patients – if this is indeed the main argument of the authors.

3. Supplementary Figure 1: It is said: : Unsupervised principal component analysis (PCA) showed that holoclones are bundled in a rather homogenous cluster (Fig. 1b, red dots) and that they are very similar amongst the 6 strains analyzed” How can this be concluded this from sup. fig1a? Green colony K38 is everywhere. only red and purple seem to be clustered to an extent?

4. Line 108+9. Based on what were these genes selected/chosen?

5. Line 362; I assume you mean the revers? FOXM1 depletion .. etc.

6. Fig. 2b. Left and right reversed in figure legend?

Reviewer #3:

Remarks to the Author:

The utility of keratinocyte skin grafts for regeneration of epidermis in burn patients is a well-established practice, and more recently, its application in a limited number of patients with junctional epidermolysis bullosa (JEB), a devastating epidermal fragility disorder, has been explored. In the latter case, a limiting factor in JEB has been the presence of few, if any, holoclones, a self-renewing stem cell population that gives rise to meroclones and paraclones eventually leading to epidermal differentiation and keratinization of the grafts.

This manuscript by Enzo et al., is an intriguing technical tour-de-force uncovering FOXM1 as a key regulator of the presence of the holocones among cultured human keratinocytes. Initially, analysis of differentially expressed genes by whole transcriptome RNA-Seq identified 526 genes which were

upregulated in holoclones when compared to either meroclonal or paraclonal. In the same analysis, 552 genes were down regulated in holoclones as compared to paraclones. The composite overexpression of 526 genes was defined as "holoclone signature". In this collection of genes within the holoclones signature, the investigators "fastened" on FOXM1, a transcription factor and member of the forkhead box family. This gene was an obvious choice, as it has been previously shown to control self-renewal of neural and hematopoietic stem cells, contribute to the regeneration of striate muscle, and has an impact on long-term maintenance of bronchiolar epithelium. However, focus on this single gene raises the question of the other 525 genes within the holoclone signature. Are any of them potential candidate genes for the holoclone maintenance? How about the genes that were down regulated in holoclones? Nevertheless, the authors make a compelling argument for FOXM1 as a key regulator of holoclones by demonstration that this gene is expressed virtually only in holoclones, its ablation by shRNA causes disappearance of the holoclone cell population, while overexpression of FOXM1 prevents the conversion of holoclones to paraclones, thus sustaining the presence of holoclones. Nevertheless, a comment on the potential role of the other 525 genes would be in order.

In order to test the role FOXM1, the investigators utilized keratinocyte cultures established from skin of patients with junctional EB which have been shown to lack the expression of the LAMB3 gene and devoid of holoclones. This study demonstrated that overexpression of FOXM1 preserved and increased the presence of holoclones in JEB cell cultures. It should be noted that while the authors state that "...enforced FOXM1 recapitulates LAMB3-mediated gene therapy of JEB...", the presence of holoclones does not result in improvement of the adhesiveness of keratinocyte grafts by these cells with biallelic nonsense mutations. It is clear that the fragility of skin in these patients can be enhanced only by genetic correction of the mutation(s) in the underlying gene (LAMB3), which could be potentially combined with FOXM1-driven increase in the presence of holoclonal stem cells.

Collectively, the data provided by Enzo et al., significantly contribute to our progress towards understanding the human epidermal biology in general, and specifically to development of gene therapy approaches for heritable skin fragility disorders, such as epidermolysis bullosa, a currently intractable group of disorders.

Reviewer #4:

Remarks to the Author:

In this article by Enzo and Seconetti et al. entitled "Single-Keratinocyte transcriptional profiling uncovers Foxm1 as a key regulator of human epidermal stem cells". The authors utilize old and new omics approaches to investigate an in vitro phenomenon called 'holoclonal'. Holoclonal are cell culture 'colonies' that are thought to arise from a single cell that gives rise to the largest type of clones compared to mera and para clones. They may represent human interfollicular epidermal stem cells. Most importantly, the physiological relevance to epidermal preparations for skin grafts, is that they are hypothesized to be required for generating colonies once grafted onto skin of burn patients and individuals suffering from EB.

Here the authors performed micro-arrays on different clones and subsequently performed single-cell-RNA-seq analysis on cultured human epidermal cells. The microarray suggested that clones derived from holoclones were unique in PC components analysis, while single-cell RNA-seq revealed unique clusters that represented holoclones. Additional single-cell RNA seq analysis revealed that holoclone clusters could be changed by inducing cultures to differentiate. Through the analysis the authors identified Foxm1 was expressed in holoclones. After which the manuscript tested whether Foxm1 could regulate the formation of holoclone in culture by manipulating gene expression through shRNAs. Finally, the authors test if Foxm1 manipulation increase holoclone formation in epidermal cells isolated from patients with EB/JEB. Foxm1 expression in EB/JEB increase holoclone formation, an important factor in the potential treatment of individuals with skin grafts for EB/JEB.

The manuscript is well written and the single-cell RNA-seq experiments are noteworthy and will provide the field with an important resource. The experiments utilizing cells from EB/JEB were

encouraging and are significant and will have impact on the field of epidermal stem cell research. The data supports the conclusions, but additional analysis such as RNA-velocity would increase the manuscripts impact but is not necessary for publication.

Reviewer #5:

Remarks to the Author:

Enzo et al. analyze properties of human epidermal stem cells through clonal assays and single-cell sequencing. The study reveals a graded distribution of stemness by clonal markers, and suggests FOXM1 as a key regulator of stem cell function. The experiments are interesting and overall well-conducted, though I have several questions regarding the analysis and presentation of results. I also have a question about how this work fits into the existing literature. These are given below.

Major points

1. In Fig 2, it is unclear how the five keratinocyte clusters were identified as H, M, P, TD1, TD2. Was this done by prior known markers? Which? Is there significance to there being two TD clusters compared to only one each of the proliferative populations? I.e. what distinguishes TD1 and TD2? In addition, can the authors speculate on why H lies closer to TD1 than to P at both t1 and t2? This seems to imply some similarity which does not make sense (taking into account the limitations of the UMAP projection, of course). This effect is also observed in Fig. 3A - a few TD cells cluster with holoclone cells.

2. Given the emphasis of single-cell profiling of epidermal stem cells, very little discussion is given on how homogeneous the holoclone population is. Indeed, in Fig 3A, the holoclone/meroclone/paraclone cells look decidedly mixed near the start of pseudotime. How homo/heterogeneous is the holoclone cluster? E.g. with respect to its marker genes? With respect to FOXM1? How does this influence the downstream results?

3. The justification given for following up on only one holoclone marker (FOXM1) is not entirely clear. Was it chosen simply based on its involvement in other proliferative processes? Given that the holoclone signature contained over 500 genes specific to that population, what is the justification for only targeting one for follow up? The expression of FOXM1 in the single-cell data seems to be relatively low in many cells (a few copies per cell). Thus it would be expected to exhibit high levels of transcriptional noise (Fig 3c) - this could obscure its expression profiles across clones. NB I *think* it is expressed at only a few copies per cell but this needs to be confirmed by adding the units to expression values in Fig. 3C - raw counts? Or normalized how?

4. Line 225 Methods not clear (scFOXM1#1, etc).

“Strikingly, ablation of FOXM1 induced the selective disappearance (strains K5 and K71) or a decrease (K52) of holoclones without altering the presence and proportion of meroclones and paraclones” - please change the language for clarity. How can the holoclones be removed without altering meroclone/paraclone %?

5. Are there any differences in YAP expression in the scRNA seq data? How about expression of YAP target genes - such presentation provide indirect estimation of nuclear YAP activity, and add support at the single-cell level to the identified interactions between FOXM1 and the YAP signaling pathways in epidermal regeneration.

6. “Of note, FOXM1 was virtually undetectable in skin sections prepared 268 from JEB-1, a homozygous carrier of a c.1954delG mutation in the LAMB3 gene (Fig. 6a).” - see comment (2) above... FOXM1 Expression is (I think) very low overall, so isn't this expected?

7. Relevance to the literature. In relation to ref. 40 the authors state: "similar clusters of basal keratinocytes have been identified in both conditions" - but do not discuss in any detail the similarities or differences between the signatures identified by the previous vs the current study. Such a comparison would allow the reader to understand the current findings in light of the literature. In addition, a number of other papers have been published describing the transcriptomes of epidermal stem cells in mouse (e.g. PMIDs: 32187560, 30332640, 27641957, 33116143). How does the current work fit into/add to what is understood about epidermal stem cell signatures from these previous studies?

REVIEWER COMMENTS

Reviewer #1 (Remarks to the Author):

This is very interesting manuscript, where the authors have shown that FOXM1 acts as regulator of human epidermal stem cells by using different approaches including clonogenic assays, genome wide transcriptomic analysis and single cell RNA sequencing, functional assays using gain and loss of function. The authors demonstrated the role of FOXM1 and its regulation by YAP in sustaining human epidermal stem cell fate. This is a very thorough and well executed study that provide important insights into the molecular mechanisms regulating human epidermal stem cells. The study is well controlled and very convincing. I recommend publication of this study to nature communications.

Comments

It will be interesting to compare the molecular features of holoclones based on bulk and single cell transcriptomic with the various transcriptomic data (populations and single cell RNA-seq) that have been published in mouse epidermal stem cells to define common epidermal stem cell and progenitor markers across species. There are various studies that have provided such bulk and single cell RNA-seq of mouse epidermal stem cells and progenitors during homeostasis, development and tissue repair (Mascre et al., 2012, Sánchez-Danés et al., Nature 2016, Joost Cell Systems 2016, Joost Cell Reports 2018, Dekoninck et al., Cell 2020, Aragona et al., Nature2020). It will be interesting to correlate holoclone and paraclone specific markers of human epidermal stem cells with mouse stem and progenitor cells.

Authors' reply:

We thank the reviewer for positive comments.

We thank the reviewer for this suggestion and added a new paragraph in the Discussion. We were aware of single-cell transcriptomic analysis of mouse skin. The only reasons why we did not initially compare those analyses with ours are that (i) murine data were generated from resting (in vivo) skin while our analysis was performed on primary epidermal cultures that mimics a wound healing scenario; (ii) our data were generated in defined human keratinocyte clonal types (H/M/P), which are not identified in murine cultures; (iii) at variance with human skin, murine skin is highly enriched in hair follicles and some signalling pathways behave differently in human and mouse skin.

Nevertheless, the point raised by the reviewer is well taken and gave us the opportunity to highlight that both human holoclone-forming cells and the population of K14+ cells containing murine epidermal stem cells (Mascre et al, 2012, Fig. 4a) upregulate genes regulating DNA repair, cell cycle, chromosome segregation. Our holoclone signature is enriched in the proliferative cluster of the murine stem cell/progenitor compartment (Dekoninck et al., 2020; Aragona et al ,2020). Human H cluster (Fig. 3a,b) and murine proliferative/undifferentiated basal stem clusters (Aragona et al, 2020) are marked by high expression of DIAPH3 and have been both identified as the starting point of the differentiation trajectory passing through the transient amplifying progenitors and then differentiated cells.

Reviewer #2 (Remarks to the Author):

The manuscript has a refreshing experimental set-up and the strategy to discriminate the different stem cell pools prior to terminally differentiated keratinocytes is well-designed. A potential new signature for holoclones would be welcome, as to be better able to serve those in need of skin transplants.

My main concerns therefore do not lie in the potential impact, nor in the execution of the study. Instead, they much more regard the true meaning and value of the FOXM1-G2/M signature in being unique to, and thus useful to detect, holoclones vs. paraclones and + terminally differentiated keratinocytes (and to a lesser extent meroclones). Holoclones divide. More YAP1 means more FOXM1 and therefore more division. But this still only confirms FOXM1 is a cell cycle gene; not that FOXM1 is unique to holoclones. Promoting proliferation of holoclones would of course be of great value, but this per sé would then be the main difference between holoclones and the other groups: G2/M status.

More specifically:

1. FOXM1 is a well-known G2/M protein. It regulates G2/M transition, amongst others, through regulation of AuroraB kinase and cyclins as CCNA (see fig. 1+2 as target genes also picked up here). The fact that all of these are elevated in the holoclone signature would be in line with the fact that it is the holoclone cells that are still proliferating and are going through G2/M phase of the cell cycle. A BrdU/EdU staining would then also likely be able to discriminate holoclones from paraclones (and to a lesser extent meroclones). To rule out we are simply looking at proliferation (G2/M status), a BrdU/EdU incorporation could easily be performed, and/or a cell cycle distribution assay by cytometry.

I suspect the authors are looking at proliferation (G2/M) markers, as the holoclones are able to divide. But that signature is not unique to holoclones (as all dividing cells will have it) and it would then also not be a specific gene signature that can be used to discriminate these cells in a complex mixture in vivo/ex vivo. In other words, can this signature be used to detect holoclones in vivo? E.g. after injury?

Authors' reply:

We thank the reviewer for positive comments.

We added more data and Figures to address the main issue raised by the reviewer (see new Suppl. Figs 1, 3 and 4) and a new paragraph has been included in the Discussion.

As requested by the reviewer, we analysed the effect of FOXM1 on keratinocyte proliferation. Neither ablation nor overexpression of FOXM1 affect the cell cycle phase distribution of clonogenic keratinocytes (new Suppl. Fig. 3, panel f and new Suppl. Fig. 4, panel d). Ki67 expression was not affected by FOXM1 overexpression (not shown). Enforced FOXM1 did not affect growth rate and number of cells generated by serial cultivation (new Suppl. Fig. 4, panel e).

In agreement with these data, clustering identified 9 cell types within the integrated data regressed for cell cycle (see text). Clusters made by holoclone- and meroclone-forming cells are comparable in terms of percentage of cells in G2/M and S phases. We are enclosing these data for the reviewer.

[Redacted]

Thus, the only clear biological effect of FOXM1 ablation was the disappearance of holoclone forming cells, while enforced FOXM1 halted clonal conversion and maintained holoclones for longer time. In fact, while the entire human epidermal basal layer is endowed with proliferative capacity, nuclear FOXM1 in vivo is detected only in few basal cells (see Fig. 6 panel a). Since FOXM1 strikingly distinguish holoclones from both meroclones and paraclones in clonal analysis (see Fig. 4, panel a), it is conceivable to speculate that those FOXM1-positive cells present in the epidermal basal layer in vivo are the ones generating holoclones in culture.

These features are in fact consistent with what has been observed in human keratinocyte cultures. All clonogenic epidermal keratinocytes are endowed with high proliferative capacity. Both holoclones and meroclones can undergo dozens of population doublings before the onset of replicative senescence (De Rosa et al., 2019). Even paraclones can undergo up to 15 cell doublings before senescence. As shown in Suppl. Fig. 1 (panel a), holoclone- and meroclone-forming cells generate colonies of the same size and the growth rate of holoclones and meroclones is virtually indistinguishable; they both reach sub-confluence in 8-10 days through an identical number of cell doublings. In fact, proliferation per se does not differ in clonogenic human keratinocytes and the only feature distinguishing holoclone-forming cells from other clonogenic keratinocytes is its self-renewal capacity (Hirsch et al., 2017), which is the main hallmark of somatic stem cells. Yet, the holoclone signature in general, and FOXM1 in particular, distinguish holoclones from other keratinocytes, including meroclones and paraclones.

2. In line with this. I suspect it is the culture media that dictates the holoclones to develop the FOXM1-G2/M signature. Would this signature still remain in holoclones that are placed on low growth factor media?

Authors' reply:

Defined culture media and feeder layer are both required to maintain all clonogenic cells including holoclones. Plating primary epidermal keratinocytes in the absence of defined additives (including growth factor, see Methods) and feeder-layer, are detrimental for the entire culture and all clonogenic cells would rapidly undergo terminal differentiation.

3. FOXM1b and c differ in ERK phosphorylation sites – which are cell cycle controlled. Of note. Also FOXM1c is expressed in cancer cells, e.g melanoma.

Does FOXM1b expression give the same phenotype in Fig. 4? This would also to an extent address the concern of FOXM1 here being merely a proliferation factor.

Authors' reply:

Isoform B is not present in human primary keratinocytes (see Suppl. Fig. 4a). To further clarify isoforms contribution to the suggested role of FOXM1 in maintaining self-renewal, we tested the effect of U0126, a MEK specific inhibitor that blocks ERK phosphorylation, on FOXM1 levels (Ma et al, 2004). ERK inhibition induces drastic reduction of FOXM1 levels, confirming that the only responsive endogenous protein is the isoform C (new Suppl. Fig. 4c). Moreover, to confirm the specific activity of FOXM1-C with respect to FOXM1-B, we overexpress both isoforms in human primary keratinocytes. As shown in new Fig. 4e, only isoform C is able to upregulate both survivin and p63.

4. Stable expression of FOXM1 would increase the change of cancer. This would impair the use of ectopic FOXM1 (and LAMB3/YAP1) expression to counteract JEB. However, there are transient ways of enforcing these proteins. Though arguably not the best when wanting to use these clones for transplantation purposes, proof-of-concept could for instance be obtained with Nicotine, which has been claimed to promote FOXM1 expression in keratinocytes. Activators of YAP/TAZ can also be explored to study the importance of FOXM1 in holoclone definition.

Authors' reply:

We thank the reviewer for the suggestion but our purpose was not to use exogeneous FOXM1 (or YAP) for clinical application. From one side, our findings shed light on the well know stem cell depletion observed during the clinical course of JEB patient, from the other side improved criteria for measuring stem cells in epidermal cultures, which is an essential feature of the transgenic graft. Genetic correction of LAMB3 and restoration of the keratinocyte adhesion machinery restore YAP function (De Rosa et al., 2019) and FOXM1 expression (this paper), hence an appropriate number of holoclone-forming cells. Thus, FOXM1 (together with p63 and YAP) would be a critical parameter to assess the quality of transgenic cultures destined to clinical application.

5. JEB-LAMB3 seems to provide much larger/stable colonies than JEB-FOXM1. Would it not be better for the authors to investigate transient LAMB3 expression in promoting holoclone abundance?

Authors' reply:

The reviewer is correct. Indeed, we observed higher colony size and increased nuclear YAP when primary JEB keratinocytes were plated onto laminin-332 coated vessels (De Rosa et al., 2019). In fact, to minimize stem cell loss potentially occurring in the initial phases of JEB keratinocyte cultivation, we are investigating the use of laminin-332-coated vessels to first establish a JEB culture. However, this cannot substitute for a stable long-term restoration of adhesion-dependent stem cells, which can be attained only by permanent expression of LAMB3.

6. In line with this, in fig. 7 it was not shown whether FOXM1 was still elevated in the JEB-YAP an dJEB-LAMB3 clones, making it a bit hard to conclude this si all caused by FOXM1, instead of other actions from LAMB3

Authors' reply:

That JEB-LAMB3 and JEB-YAP contains FOXM1 (at levels comparable to normal control) is shown in Fig. 6 (panels b and c). We now added a new panel (e) to Figure 6 showing the restoration of FOXM1 expression upon LAMB3 or YAP transduction.

We agree with reviewer. YAP/FOXM1 effects on epidermal stem cells require a proper adhesion of basal cell to the basal lamina (hence a proper assembly of laminin-332). Both enforced YAP (De Rosa et al., 2019) and FOXM1 (this paper) can rescue stem cells but, obviously, do not rescue adhesion (Fig. 7). Thus we cannot exclude, actually it is conceivable that LAMB3 gene correction could have other effects and YAP/FOXM1 activation, though critical for epidermal stem cell maintenance, is part of them.

Textual / individual results:

1. Why do the authors group telomerase with the DNA-repair genes?

Authors' reply:

The reviewer is correct. Telomerase is better related to stability. In any case, we modified the sentence in discussion.

2. It would be good to include a concluding remark in the abstract on the translational potential of the findings of FOXM1 expression for JEB patients – if this is indeed the main argument of the authors.

Authors' reply:

We thank the reviewer for the suggestion, a sentence has been added in the abstract.

3. Supplementary Figure 1: It is said: "Unsupervised principal component analysis (PCA) showed that holoclones are bundled in a rather homogenous cluster (Fig. 1b, red dots) and that they are very similar amongst the 6 strains analyzed" How can this be concluded this from sup. fig1a? Green colony K38 is everywhere. only red and purple seem to be clustered to an extent?

Authors' reply:

The purpose of Supplementary Fig 1b is to show that no homogeneous cluster could be defined based only on strains, as highlighted by the reviewer. We clarified this concept in the main text.

4. Line 108+9. Based on what were these genes selected/chosen?

Authors' reply:

We have selected some of the more differentially expressed genes that were not simply related to the cell cycle. This has been clarified in the text.

5. Line 362; I assume you mean the revers? FOXM1 depletion .. etc.

Authors: We thank the reviewer, we fixed this mistake.

6. Fig. 2b. Left and right reversed in figure legend?

Authors: We thank the reviewer, we fixed this mistake.

Reviewer #3 (Remarks to the Author):

The utility of keratinocyte skin grafts for regeneration of epidermis in burn patients is a well-established practice, and more recently, its application in a limited number of patients with junctional epidermolysis bullosa (JEB), a devastating epidermal fragility disorder, has been explored. In the latter case, a limiting factor in JEB has been the presence of few, if any, holoclones, a self-renewing stem cell population that gives rise to meroclones and paraclones eventually leading to epidermal differentiation and keratinization of the grafts. This manuscript by Enzo et al., is an intriguing technical tour-de-force uncovering FOXM1 as a key regulator of the presence of the holocones among cultured human keratinocytes. Initially, analysis of differentially expressed genes by

whole transcriptome RNA-Seq identified 526 genes which were upregulated in holoclones when compared to either meroclones or paraclones. In the same analysis, 552 genes were down regulated in holoclones as compared to paraclones. The composite overexpression of 526 genes was defined as “holoclone signature”. In this collection of genes within the holoclones signature, the investigators “fastened” on FOXM1, a transcription factor and member of the forkhead box family. This gene was an obvious choice, as it has been previously shown to control self-renewal of neural and hematopoietic stem cells, contribute to the regeneration of striate muscle, and has an impact on long-term maintenance of bronchiolar epithelium. However, focus on this single gene raises the question of the other 525 genes within the holoclone signature. Are any of them potential candidate genes for the holoclone maintenance? How about the genes that were down regulated in holoclones? Nevertheless, the authors make a compelling argument for FOXM1 as a key regulator of holoclones by demonstration that this gene is expressed virtually only in holoclones, its ablation by shRNA causes disappearance of the holoclone cell population, while overexpression of FOXM1 prevents the conversion of holoclones to paraclones, thus sustaining the presence of holoclones. Nevertheless, a comment on the potential role of the other 525 genes would be in order. In order to test the role FOXM1, the investigators utilized keratinocyte cultures established from skin of patients with junctional EB which have been shown to lack the expression of the LAMB3 gene and be devoid of holoclones. This study demonstrated that overexpression of FOXM1 preserved and increased the presence of holoclones in JEB cell cultures. It should be noted that while the authors state that “...enforced FOXM1 recapitulates LAMB3-mediated gene therapy of JEB...”, the presence of holoclones does not result in improvement of the adhesiveness of keratinocyte grafts by these cells with biallelic nonsense mutations. It is clear that the fragility of skin in these patients can be enhanced only by genetic correction of the mutation(s) in the underlying gene (LAMB3), which could be potentially combined with FOXM1-driven increase in the presence of holoclonal stem cells. Collectively, the data provided by Enzo et al., significantly contribute to our progress towards understanding the human epidermal biology in general, and specifically to development of gene therapy approaches for heritable skin fragility disorders, such as epidermolysis bullosa, a currently intractable group of disorders.

Authors' reply:

We thank the reviewer for the positive comments. The reviewer highlights important questions, as the role of at least some of the other genes, which will be addressed in further investigations. The reviewer is absolutely correct: we did not mean that FOXM1 could substitute for LAMB3 gene therapy. Enforced FOXM1 (and YAP) recapitulates LAMB3 gene therapy only for stem cell rescue. This sentence has been modified and made more clear.

Reviewer #4 (Remarks to the Author):

In this article by Enzo and Seconetti et al. entitled “Single-Keratinocyte transcriptional profiling uncovers Foxm1 as a key regulator of human epidermal stem cells”. The authors utilize old and new omics approaches to investigate an in vitro phenomenon called ‘holoclones’. Holoclones are cell culture ‘colonies’ that are thought to arise from a single cell that gives rise to the largest type of clones compared to mera and para clones. They may represent human interfollicular epidermal stem cells. Most importantly, the physiological relevance to epidermal preparations for skin grafts, is that they are hypothesized to be required for generating colonies once grafted onto skin of burn patients and individuals suffering from EB. Here the authors performed micro-arrays on different clones and subsequently performed single-cell-RNA-seq analysis on cultured

human epidermal cells. The microarray suggested that clones derived from holoclones were unique in PC components analysis, while single-cell RNA-seq revealed unique clusters that represented holoclones. Additional single-cell RNA seq analysis revealed that holoclone clusters could be changed by inducing cultures to differentiate. Through the analysis the authors identified Foxm1 was expressed in holoclones. After which the manuscript tested whether Foxm1 could regulate the formation of holoclone in culture by manipulating gene expression through shRNAs. Finally, the authors test if Foxm1 manipulation increase holoclone formation in epidermal cells isolated from patients with EB/JEB. Foxm1 expression in EB/JEB increase holoclone formation, an important factor in the potential treatment of individuals with skin grafts for EB/JEB. The manuscript is well written and the single-cell RNA-seq experiments are noteworthy and will provide the field with an important resource. The experiments utilizing cells from EB/JEB were encouraging and are significant and will have impact on the field of epidermal stem cell research. The data supports the conclusions, but additional analysis such as RNA-velocity would increase the manuscripts impact but is not necessary for publication.

Authors' reply:

We thank the reviewer for the positive comments.

Reviewer #5 (Remarks to the Author):

Enzo et al. analyze properties of human epidermal stem cells through clonal assays and single-cell sequencing. The study reveals a graded distribution of stemness by clonal markers, and suggests FOXM1 as a key regulator of stem cell function. The experiments are interesting and overall well-conducted, though I have several questions regarding the analysis and presentation of results. I also have a question about how this work fits into the existing literature. These are given below.

Major points

1. In Fig 2, it is unclear how the five keratinocyte clusters were identified as H, M, P, TD1, TD2. Was this done by prior known markers? Which?

Authors' reply:

The cluster identification was unsupervised; the algorithm itself identified 5 clusters which were labelled by the authors afterwards. Labels were assigned based on some already known markers (fig. 2c in main text – clonogenic markers and differentiation markers) and on differentially expressed genes that were identified through microarrays experiments (fig. 2c in main text – holoclone markers).

We tried to better clarify this and re-phrased the entire paragraph in the main text.

Is there significance to there being two TD clusters compared to only one each of the proliferative populations? I.e. what distinguishes TD1 and TD2? In addition, can the authors speculate on why H lies closer to TD1 than to P at both t1 and t2? This seems to imply some similarity which does not make sense (taking into account the limitations of the UMAP projection, of course). This effect is also observed in Fig. 3A - a few TD cells cluster with holoclone cells.

Authors' reply:

TD1 and TD2 are clusters made by non clonogenic cells, with TD2 being more differentiated than TD1 (note expression of TGM1, SPINK5 and IVL in Fig. 2c). Disposition of clusters in UMAP projection in fig. 2a is biased by the presence of fibroblasts that lead to a compression of keratinocytes clusters on the right of the graph, due to the very different transcriptional profile. Taking into account the limitation of 2D projection also for the UMAP in Fig. 3a, the presence of some TD1 close to H could have been influenced both by the presence of doublets (a terminally differentiated and a clonogenic cell in the same droplet) and by their proliferation state.

2. Given the emphasis of single-cell profiling of epidermal stem cells, very little discussion is given on how homogeneous the holoclone population is. Indeed, in Fig 3A, the holoclone/meroclone/paraclone cells look decidedly mixed near the start of pseudotime. How homo/heterogeneous is the holoclone cluster? E.g. with respect to its marker genes? With respect to FOXM1? How does this influence the downstream results?

Authors' reply:

The transition from keratinocyte stem cells (holoclone forming cells) to transient progenitors (meroclone and paraclone forming cells) is a continuous biological process. Therefore, it was not expected a neat division between the clusters along the pseudotime. For the same reason, it is important to consider the relative variations in the expression of marker genes (see for instance p63, but also YAP and, to a lesser extent FOXM1) and not simply their presence/absence. To date, we did not observe obvious differences amongst holoclones in terms, for instance, of expression of FOXM1, nuclear YAP or p63, but this notion would not be sufficient to rule out differences amongst holoclone-forming cells.

This said, the reviewer's comment is important and stimulating. Indeed, how homogeneous is (not only from a transcriptional but also from a functional point of view) the population of holoclone-forming cells is a quite important question, which might even affect further refinements in procedures aimed at their clinical application. We are dealing with human cells, thus we cannot take advantage of in vivo type of experiments. In any case, these clonal types can be identified in many mammals (for instance pigs) but not in mice. Analysing primary clonogenic keratinocytes at clonal level is already a cumbersome, time-consuming experiment to set up (lots of cell culture is involved) and one way to address this important issue is to perform clonal analysis of the clones (a double clonal analysis blindly performed). We are actually planning these types of experiments.

3. The justification given for following up on only one holoclone marker (FOXM1) is not entirely clear. Was it chosen simply based on its involvement in other proliferative processes? Given that the holoclone signature contained over 500 genes specific to that population, what is the justification for only targeting one for follow up?

Authors' reply:

Being somehow undoable to deeply and contemporary analyse many different genes, we initially focused our attention on FOXM1 mainly because FOXM1 pathway is amongst the gene sets enriched in holoclones both in microarray (Fig. 1f) and scRNA-seq data (Supplementary Fig. 2e). Moreover it has been reported to be an important gene involved in stem cell biology of several tissues, i.e. embryonic stem cells, neural and hematopoietic stem cells, striate muscle and bronchiolar epithelium. We tried to better clarify this choice in the main text.

The expression of FOXM1 in the single-cell data seems to be relatively low in many cells (a few copies per cell). Thus it would be expected to exhibit high levels of transcriptional noise (Fig 3c) - this could obscure its expression profiles across clones. NB I *think* it is expressed at only a few copies per cell but this needs to be confirmed by adding the units to expression values in Fig. 3C - raw counts? Or normalized how?

Authors' reply:

The most representative transcripts in the dataset are keratins (7%). Being FOXM1 a transcription factor, its activity needs to be strictly tuned. In fact, other transcription factors such as TP63 have a comparable number of transcripts per cell. Despite these

biological requirements, the H cluster has a higher percentage of cells carrying at least one copy of FOXM1 transcript (68.9%) in comparison with M (47.9%) and P (16.1%). In addition, in fig. 2c we can observe that in H the average expression of FOXM1 among the cluster is higher than in M and P. In Fig. 3c, y axes of kinetics plots represent expression values normalized and rounded by Monocle3. These plots were generated with the plot_genes_in_pseudotime function.

4. Line 225 Methods not clear (scFOXM1#1, etc).

“Strikingly, ablation of FOXM1 induced the selective disappearance (strains K5 and K71) or a decrease (K52) of holoclones without altering the presence and proportion of meroclones and paraclones” - please change the language for clarity. How can the holoclones be removed without altering meroclone/paraclone %?

Authors' reply:

We thank the reviewer for the suggestion, we clarified the concept as follows: “Strikingly, ablation of FOXM1 induced the selective disappearance (strains K5 and K71) or a decrease (K52) of holoclones. Relative amount of meroclones and paraclones was not significantly altered (Fig. 4c, left panel)”.

Please, consider that clonal conversion, i.e. the transition from holoclones to meroclones and paraclones, is a continuous and progressive biological process and there are variations on the relative percentage of meroclones and paraclones, depending upon the donor, the age of the donor, the number of cell passages. More importantly, holoclones represents only approximately 5% of all clonogenic cells. Thus, a total disappearance of the population of holoclone forming cells in the absence of significant variation of the relative amount of meroclones and paraclones is not surprising in this biological system.

5. Are there any differences in YAP expression in the scRNA seq data? How about expression of YAP target genes - such presentation provide indirect estimation of nuclear YAP activity, and add support at the single-cell level to the identified interactions between FOXM1 and the YAP signaling pathways in epidermal regeneration.

Authors' reply:

As shown in the Figure below, YAP and TAZ (WWTR1) expression do not vary among clusters in scRNA-seq data. This was expected because YAP activity is not regulated by its expression but by its localization in the cell (Dupont 2010, De Rosa, 2019), which depends on YAP phosphorylation. Indeed, GSEA analysis from microarray and single cell data highlighted that YAP/TAZ activity (not expression) is upregulated in holoclones and holoclone-forming cells.

[Redacted]

6. “Of note, FOXM1 was virtually undetectable in skin sections prepared 268 from JEB-1, a homozygous carrier of a c.1954delG mutation in the LAMB3 gene (Fig. 6a).” - see comment (2) above... FOXM1 Expression is (I think) very low overall, so isn't this expected?

Authors' reply:

In vivo, stem cells are interspersed in the basal layer among the other clonogenic cells. Therefore, FOXM1 positive cells follow this pattern. Absence of FOXM1 in skin biopsy derived from JEB-1 is coherent with the depletion of stem cells due to the pathology.

7. Relevance to the literature. In relation to ref. 40 the authors state: “similar clusters of basal keratinocytes have been identified in both conditions” - but do not discuss in any detail the similarities or differences between the signatures identified by the previous vs the current study. Such a comparison would allow the reader to understand the current findings in light of the literature.

Authors' reply:

We thank the reviewer for this suggestion and added in the discussion the following sentence: “Despite this fundamental difference, similar clusters of basal keratinocytes have been identified in both conditions: the BAS-II cluster is similar to our H cluster and the BAS-I cluster is similar to our M cluster (Wang et al, 2020). The notion that the BAS-II cluster contains more cells than BAS-I is consistent with the notion that, as opposed to primary cultures, single clonogenic keratinocytes directly cultivated from a skin biopsy generate many more holoclones than meroclones (De Rosa et al, 2020)”

In addition, a number of other papers have been published describing the transcriptomes of epidermal stem cells in mouse (e.g. PMIDs: 32187560, 30332640, 27641957, 33116143). How does the current work fit into/add to what is understood about epidermal stem cell signatures from these previous studies?

Authors' reply:

We thank the reviewer for this suggestion and added a new paragraph in the Discussion.

The only reasons why we did not initially compare murine analyses with ours are that (i) murine data were generated from resting (in vivo) skin while our analysis was performed on primary epidermal cultures that mimicks a wound healing scenario; (ii) our data were generated in defined human keratinocyte clonal types (H/M/P), which are not identified in murine cultures; (iii) at variance with human skin, murine skin is highly enriched in hair follicles and some signalling pathways behave differently in human and mouse skin.

Reviewers' Comments:

Reviewer #2:

Remarks to the Author:

I remain of the opinion that the study design is very nice. However, as much as this deserves praise, I remain concerned that the main argument from point 1 still stands.

a. The supplementary figures are informative. However, Sup. Fig. 1c left panel then proves the point I raised, right? There is a large fold enrichment in cell cycle, chromosome segregation and spindle organization between holoclones vs. meroclones (and even more vs. paraclones). This was exactly my point. Holoclones are able to divide more and thus have these pathways, which are all regulated by FOXM1, elevated. So, as per my original concern: "More YAP1 means more FOXM1 and therefore more division. But this still only confirms FOXM1 is a cell cycle gene; not that FOXM1 is unique to holoclones." Did you not exactly prove the point with this figure?

b. I do think the reviewer figure you provide is very informative. Given the fact that probably more scientists familiar with FOXM1 regulation of G2/M will have similar questions, I suggest including this figure from the reviewer answers in the main text. That dais, the cell cycle distribution is very different from that in Sup. Fig 3f. So, this suggests that cell cycle regulation and G2/M enrichment by FOXM1 does indeed play a major role in Holo- and meroclone maintenance. In this regard it would help to have a similar plot for the clones in Fig. 4C and 4F. Do the holo vs. meroclones of, for instance, K5, K52, K71, K38 and K49 also show equal G2/M distribution?

c. As much as this reviewer figure is useful, it still only shows that the clones are in a certain state, not whether they actually divide similarly. As such, I had suggested an EdU/BrdU incorporation, or a Ki67 staining. Why has this not been performed? Such an assay would lay this discussion to rest. Do meroclones show reduced EdU incorporation, despite having a similar cell cycle distribution? If so, the authors showed the FOXM1 is a

d. Sup. Fig. 4 does not solve the matter much, since this is in keratinocytes and not in the holo-meroclones, correct?

Reviewer #5:

Remarks to the Author:

The authors have clarified and revised the manuscript in response to the questions raised. It is, I believe, much improved as a result, and I now find the results easier to follow and presented more clearly.

A couple important questions/clarifications remain.

1. Some ambiguity remains regarding the identification of keratinocyte clusters - it would be helpful to include somewhere in the Results section a statement to the effect of: the clusters names ("H, M, P, TD1, and TD2") were assigned based on the Holocene signature and previously known markers. In addition, for clarity, statements such as

"Clusters TD1 and TD2 (23,4% and 11%, respectively)" could be rephrased as "the clusters that we designate as TD1 and TD2, based on their expression of XYZ...."

2. Although in the response letter the method used to generate Fig 3C is given (from within Monocle 3), the description in the legend of Fig 3 still does not contain the necessary information of what normalization was performed on these genes.

REPLY TO REVIEWERS' COMMENTS

Reviewer #2 (Remarks to the Author):

I remain of the opinion that the study design is very nice. However, as much as this deserves praise, I remain concerned that the main argument from point 1 still stands. a. The supplementary figures are informative. However, Sup. Fig. 1c left panel then proves the point I raised, right? There is a large fold enrichment in cell cycle, chromosome segregation and spindle organization between holoclones vs. meroclones (and even more vs. paraclones). This was exactly my point. Holoclones are able to divide more and thus have these pathways, which are all regulated by FOXM1, elevated. So, as per my original concern: "More YAP1 means more FOXM1 and therefore more division. But this still only confirms FOXM1 is a cell cycle gene; not that FOXM1 is unique to holoclones." Did you not exactly prove the point with this figure?

b. I do think the reviewer figure you provide is very informative. Given the fact that probably more scientists familiar with FOXM1 regulation of G2/M will have similar questions, I suggest including this figure from the reviewer answers in the main text. That is, the cell cycle distribution is very different from that in Sup. Fig 3f. So, this suggests that cell cycle regulation and G2/M enrichment by FOXM1 does indeed play a major role in Holo- and meroclone maintenance. In this regard it would help to have a similar plot for the clones in Fig. 4C and 4F. Do the holo vs. meroclones of, for instance, K5, K52, K71, K38 and K49 also show equal G2/M distribution?

c. As much as this reviewer figure is useful, it still only shows that the clones are in a certain state, not whether they actually divide similarly. As such, I had suggested an EdU/BrdU incorporation, or a Ki67 staining. Why has this not been performed? Such an assay would lay this discussion to rest. Do meroclones show reduced EdU incorporation, despite having a similar cell cycle distribution? If so, the authors showed the FOXM1 is a d. Sup. Fig. 4 does not solve the matter much, since this is in keratinocytes and not in the holo- meroclones, correct?

REPLY:

We added "proliferative potential" in the title and added few sentences in the text (Results and Discussion) better explaining this issue in relation to the different clonal types. We also added the cell cycle figure, originally sent to the reviewer only, as a new panel (d) in Suppl. Fig. 2.

Concerning the issue "d" raised by the reviewer: yes, the data have been produced on mass keratinocyte cultures and not on clonal keratinocyte cultures (holoclones/meroclones/paraclones) also because those experiments can be done only on clonal progeny. But the message from these experiments is that neither ablation (Suppl. Fig. 3) nor overexpression (Suppl Fig. 4) of FOXM1 have any obvious effect on cell cycle and growth rate of primary epidermal cells. On the other hand, the set of experiments shown in Fig. 4 clearly show that FOXM1 ablation abolishes holoclones but not meroclones and paraclones (Fig. 4c) and enforced FOXM1 blocks clonal conversion and sustain holoclones, without significantly altering mero and paraclones (Fig. 4f).

Just for the sake of clarity, let us better clarify the point raised by the Reviewer:

The proliferative compartment of human epidermis is quite complex and contains different types of clonogenic cells, ALL of which proliferate. They differ in terms of self-renewal and long-term proliferative potential, both of which are hallmarks of holoclone-forming cells but not of mero/paraclone-forming cells (Hirsch et al., Nature 2017).

Holoclone-forming cells are epidermal stem cells. Mero/paraclone-forming cells are transient progenitors. This notion has been definitively and unambiguously proven by clonal tracing experiments performed on transgenic epidermis in vitro and in vivo (Hirsch et al., Nature, 2017). Clonal conversion, that is the transition from holoclone to meroclone to paraclone is an unidirectional process and holoclone-forming cells account for the entire proliferative capacity of the epidermis (Mathor et al., PNAS 1996; Hirsch et al., Nature 2017).

This said, the vast majority (over 95%) of keratinocytes forming an holoclone is able to re-initiate daughter colonies able to proliferate and self-renew, whilst meroclones and paraclones are formed by keratinocytes that, although still proliferating within the colony, progressively lose their capacity to re-form new growing colonies (Pellegrini et al., J.Cell Biol., 1999). In other words, at variance with holoclones, meroclones contains cells that do not further proliferate upon passaging. The vast majority (over 95%) of keratinocytes forming a paraclone is unable to re-initiate daughter colonies. As a result of such clonal conversion, meroclones and paraclones still undergo several cell doublings before reaching replicative senescence (Pellegrini et al., 1999; De Rosa et al., Cell Rep. 2019).

That's why cell cycle genes are more expressed in the holoclone cluster.

FOXM1 (and YAP) is instrumental to the clonal conversion process, which regulate the long-term proliferative potential and the self-renewal marking specifically the holoclone-forming cells (Mathor et al., PNAS 1996; Hirsch et al., Nature 2017). Indeed, FOXM1 is highly expressed in holoclones, barely detectable in meroclones and virtually undetectable in paraclones (Fig.4a), FOXM1 ablation abolishes holoclones but not meroclones and paraclones (Fig. 4c) and enforced FOXM1 simply blocks clonal conversion (Fig. 4f).

That's why we state that YAP/FOXM1 is important for the proliferative potential and not simply for the cell cycle as such, at least on our cells.

Reviewer #5 (Remarks to the Author):

The authors have clarified and revised the manuscript in response to the questions raised. It is, I believe, much improved as a result, and I now find the results easier to follow and presented more clearly.

A couple important questions/clarifications remain.

1. Some ambiguity remains regarding the identification of keratinocyte clusters - it would be helpful to include somewhere in the Results section a statement to the effect of: the clusters names ("H, M, P, TD1, and TD2") were assigned based on the Holoclone signature and previously known markers. In addition, for clarity, statements such as "Clusters TD1 and TD2 (23,4% and 11%, respectively)" could be rephrased as "the clusters that we designate as TD1 and TD2, based on their expression of XYZ...."

2. Although in the response letter the method used to generate Fig 3C is given (from within Monocle 3), the description in the legend of Fig 3 still does not contain the necessary information of what normalization was performed on these genes.

REPLY

We thank the reviewer for the suggestions.

1. We changed the statement as follows:

“The clusters that we designate as TD1 and TD2 (23,4% and 11%, respectively) express high levels of markers of terminal differentiation, such as SERPINB3, SFN, KRT10, TGM1, IVL and SPINK5 (Fig. 2c, differentiation markers)”.

2. We added in the legend Figure 3 the information required on the method to generate panel c.